## RESEARCH ARTICLE

# Bayesian data driven modelling of kinetochore dynamics: Space-time organisation of the human metaphase plate

**Constandina Koki[1], Alessio V. Inchingolo[2], Abdullahi Daniyan[1], Enyu Li[1], Andrew D. McAinsh[2]\*, Nigel J. Burroughs[1]\***

**1** Zeeman Institute (SBIDER), Mathematics Institute, University of Warwick, Coventry, United Kingdom,
**2** Centre for Mechanochemical Cell Biology, Warwick Biomedical Sciences, Warwick Medical School, University of Warwick, Coventry, United Kingdom

\* A.D.McAinsh@warwick.ac.uk (ADM); N.J.Burroughs@warwick.ac.uk (NJB)

## Abstract

Mitosis is a complex self-organising process that achieves high fidelity separation of duplicated chromosomes into two daughter cells through capture and alignment of chromosomes to the spindle mid-plane. Chromosome movements are driven by kinetochores (KTs), multi-protein machines that attach chromosomes to microtubules (MTs), and through those attachments both control and generate directional forces. Using lattice light sheet microscopy imaging and automated near-complete tracking of kinetochores at fine spatio-temporal resolution, we produce a detailed atlas of kinetochore metaphase-anaphase dynamics in untransformed human cells (RPE1). Such data allows dynamic models to be reverse engineered and biological hypotheses to be addressed. We determined the support from this dataset for 17 models of metaphase dynamics using Bayesian inference, demonstrating (1) substantial sister asymmetry that generates transverse organisation of the metaphase plate (MPP), (2) substantial spatial organisation of KT dynamic properties within the MPP, and (3) significant time dependence of the K-fiber mechanical parameters whereby K-fiber forces tune over the last 5 mins of metaphase towards a set point, referred to as the anaphase ready state. These spatio-temporal trends are robust to perturbation of the spindle assembly pathway (nocodazole washout treatment), suggesting that the underlying processes generating kinetochore heterogeneity are intrinsic to mitosis and possibly play a role in ensuring high-fidelity segregation.

## Author summary

Cell division segregates newly duplicated chromosomes into two daughter cells. This is a mechanical process orchestrated by the mitotic spindle, a self-assembling molecular machine comprising dynamic fibres called microtubules.

**Data availability statement:** Data and Code for running the models that are presented in this manuscript can be found in GitHub: https://github.com/ckoki21/MetaAnaDynamics.

**Funding:** C.K., A.V.I, A.D. were supported by BBSRC (BB/R009503/1), and A.D., A.D.M. were supported by a Wellcome Senior Investigator Award (grant 106151/Z/14/Z). The funders had no role in study design, data collection and analysis, decision to publish, or preparation of the manuscript.

**Competing interests:** The authors have declared that no competing interests exist.

Chromosomes are attached to microtubules at protein complexes called kineto-chores, forces from these attachments driving chromosome movements. During cell division the chromosomes are aligned at the cell equator, where they undergo pseudo-periodic oscillations for ∼10 minutes before chromosome segregation begins. The purpose of this 10-minute 'holding pattern' is however unclear. By tracking kinetochores in 3D, we estimated the forces acting on individual kineto-chores using reverse engineering techniques. We discovered that kinetochore dynamics is very variable within a cell, with substantial dependence of forces on spindle location and substantial changes in forces occurring in the lead up to anaphase and segregation. We speculate that this variability is a design feature, with forces adapting to the local spindle curvature, whilst random variability may be important to prevent oscillation synchrony. This work establishes a frame-work to quantitatively analyse how kinetochores work collectively at the cell level, and how individual variability may contribute to cell division robustness. Our work demonstrates that kinetochores are not equal, a fact that may be crucial in unravelling error detection and correction mechanisms.

## 1 Introduction

Chromosome segregation relies on the self-assembly of the microtubule-based bipo-lar spindle, a dynamic structure, with the microtubules (MTs) undergoing repeated cycles of polymerisation and depolymerisation (dynamic instability), [1]. Microtubules are nucleated at the centrosomes located at each spindle pole, and extend radially from the spindle poles with subsets forming discrete bundles of ∼10 microtubules [2] that connect to each of the 92 sister chromatids (replicated chromosomes). These bundles are the K-fibers that attach to the chromatids at the kinetochores (KTs), multi-protein machines which maintain attachment to the plus-ends of microtubule bundles as they grow and shrink. These dynamic K-fibers independently generate the pushing and pulling forces that orchestrate chromosome movements, and ultimately leads to the segregation of sister chromatids into respective daughter cells during anaphase, [3,4].

Each of the 46 sister KT pairs must become bi-orientated, the attachment config-uration where sister chromatids of a pair are attached by K-fibers to opposite spin-dle poles – this is the only geometry compatible with accurate chromosome segre-gation. Concurrently with biorientation, the chromosomes congress, *i.e.,*the chromo-somes align at the cell equatorial (mid-)plane forming the metaphase plate (MPP). Over the ∼10 minute course of metaphase, chromosomes undergo quasi-periodic oscillations along the spindle axis, [5–7]. These oscillations are largely driven by sis-ter KTs switching between poleward (P; attached microtubules depolymerising) and away-from-the-pole (AP; attached microtubules polymerising) moving states. When one sister is P and the other AP, the sister chromatids undergo sustained directional movement. Directional switches arise when both sister KTs switch directional state

(P, AP), a coherent switching that is believed to be regulated by tension in the centromeric spring; centromeric chromatin connecting sister chromatids behaves as an elastic spring, [6,8]. Metaphase oscillations provide a unique opportunity to examine the mechanisms by which KTs generate and sense forces.

The purpose of metaphase remains a mystery. One possibility is that it simply reflects a "waiting" state before anaphase onset, with metaphase duration being determined by the rates of biochemical events that prepare for, and initiate, chromosome segregation in anaphase. For instance, once all KTs are correctly attached to microtubules, thereby satisfying the spindle checkpoint (SAC), Cyclin and Securin destruction is initiated. Mitotic slippage, where anaphase proceeds with one or more incorrectly attached KTs is rare, [9]. However, there are a number of observations that suggest that metaphase has a more vital role to play in mitosis. Specifically, in cells with an asymmetric centriole distribution between the two spindle poles anaphase is delayed until the MPP is centralised [10]. This is due to a failure to stabilise microtubule attachments in asymmetric spindles, thus preventing SAC satisfaction. This suggests that additional quality control processes exist in metaphase, here spindle symmetrisation control, and the duration of metaphase reflects the time to satisfy all quality requirements. There is also evidence of mechanical changes during metaphase suggesting active processes are present that continue to remodel the spindle during metaphase and modify chromosome dynamics. This includes a thinning of the MPP (reduction in the MPP width along spindle axis) thought to be related to a decrease in kinetochore speed, [7], a decrease in KT swivel (increased alignment of the intra-kinetochore axis with the sister-sister axis) reflecting a decrease in the torque acting on KTs, [11] and a synchronisation of tension across sister pairs, [12]. We refer to this body of changes as *metaphase maturation*. Of note is that metaphase oscillations in cancer cells are attenuated, [13,14], and that centromeric spring maturation from prometaphase to metaphase is disrupted in aneuploid cell lines, [15].

There is no mechanistic understanding of metaphase maturation. A key challenge is that mitotic events occur over multiple time scales from seconds to an hour. Specifically, i) directional switching (whereby both sisters switch direction) is fast with a timescale of seconds, [8], ii) quasi-periodic metaphase oscillations have a period of the order of a minute, iii) maturation of the MPP is slow with a timescale of minutes (reduction in MPP width, increased sister-sister axis alignment), iv) the spindle turns over on a ∼5 minute timescale (MT retrograde flux is 0.8-1.5 $\mu$m/min, [16], and the half-spindle size is approximately 6 $\mu$m in RPE1 cells), and v) mitosis occurs over a duration of 1 hour (nuclear envelope breakdown (NEBD) to anaphase onset is around 25 mins). In addition, chromosomes are not identical, with both variation in size and dynamic heterogeneity. Specifically, metaphase kinetochore oscillation quality varies substantially within a cell, including a fraction of non-oscillating pairs, whilst the position of the chromosome within the 3D spindle has been reported to influence mechanical forces with both Polar Ejection Forces (PEF), [17,18] and KT swivel, [11], increasing towards the periphery of the metaphase plate. Non-sister kinetochores may also influence each others' behaviour, the dynamics of neighbouring (non-sister) KTs in the MPP being correlated, [19], and hypothesised to be due to cross-linking between K-fibres, [19,20]. The biophysical properties and dynamics of KTs are also expected to vary simply because of stochasticity inherent in the spindle self-assembly process. This includes outer KT assembly and the formation of K-fibers following nuclear envelope breakdown.

Understanding this complex multi-scale mechanical system requires development of quantitative mathematical models that capture crucial characteristics of the system's biophysics and regulatory properties. Such models are then able to provide quantitative support for conceptual ideas and generate testable predictions. Efforts in this direction have been ongoing since the 1980's [21,22] with previous work focusing on microscopic models of kinetochore-microtubule attachment, [23,24], and a range of dynamic models that explore various aspects of chromosome dynamics and associated control processes, including (but not limited to) the mechanisms that generate metaphase oscillations, [18,25–28], the role of bridging fibres and generation of spindle shape [29,30], mechanisms that reproduce chromosome congression, [16,31–37], formation and stability of bipolar spindles [38,39], and the anaphase transition, [40]. Careful calibration of models to experimental data is crucial to ensure model validity. However, model parameters cannot typically be directly estimated from data because of its high dynamic complexity and the presence of multiple mechanical processes; direct

measures such as KT speed and oscillation period have a nonlinear dependence on multiple mechanical parameters and thus present a complex inverse problem. Therefore few studies have inferred model parameters directly from experimental data. In previous work, [17] we fitted a biophysical model of metaphase oscillations to 3D kinetochore tracking data from HeLa cells, a transformed human cancer cell line that exhibits aberrant chromosome copy number. The fitted model provided fundamental insight into the forces acting on kinetochores and how sister kinetochores coordinate directional switching, [8,17]. A similar analysis on non-cancer cells has not been carried out, thus given that mitosis in cancer cells is substantially perturbed, [13,15], there is a fundamental gap in our knowledge.

In this work, we generalise the paired sister kinetochore mechanical model of [17] to incorporate sister asymmetry such that sister KTs/K-fibers are not necessarily dynamically identical, introduce time dependence in model parameters (thereby capturing metaphase maturation), and extend the model from metaphase to anaphase. Using Bayesian inference we parametrised our biophysical models from experimental KT trajectory data acquired from living human cells with lattice light sheet microscopy; all model parameters were inferrable (the unidentifiable natural length of the centromeric spring was determined as in [17]). We then used model selection methods (Bayes factor assessment of model preference, [41]), to determine which models, and thus mechanisms, are supported by the data. This data-driven approach provides key insights into how sister kinetochore dynamics are defined by both spatial and temporal cues.

## 2 Results

### 2.1 Near-complete kinetochore tracking in metaphase and through the metaphase-anaphase transition

To obtain insight into chromosome dynamics in metaphase and through to anaphase, we developed a tracking algorithm (KiT v3.0, [42], based on earlier versions [43]), that achieves near-complete tracking of fluorescently-labelled KTs in human RPE1 cells [44]. The tracking pipeline consists of: deconvolving the 4D movies; detecting candidate spots with a Constant False Alarm Rate (CFAR) based spot detection algorithm, [42]; refining spot locations using a Gaussian mixture model to provide subpixel resolution; fitting a plane to the KT population thus defining the metaphase plate and an associated reference coordinate system; linking detected particles between frames over time to form tracks; and pairing sister KTs based on their metaphase dynamics. This provides sub-pixel resolution for the positions of each KT, and allows us to study the dynamics of the (near complete) complement of sister KT pairs within a cell.

We performed live-cell imaging of RPE1 cells expressing endogenous NDC80-GFP (a kinetochore marker [44]) using lattice light sheet microscopy (Fig 1). Data were collected at a high temporal resolution of 2.05s per z-stack over typically tens of minutes, starting during metaphase and extending through anaphase. Tracking performance is shown for a typical cell in Fig 1, where we detected an average of 90 spots over 350 frames (724.5 secs), with 82 KTs tracked throughout the whole movie and 43 KT pairs tracked for at least 75% of the movie (38 KT pairs were tracked for the entire duration of metaphase, see Section C1 in S1 Appendix and Fig A in S1 Appendix). This is close to the expected 92 KTs (46 paired chromatids) for a human cell line with a diploid 46,XY karyotype. This cell's KT pair tracks normal to the metaphase plate (MPP) are shown in Fig 1E. We imaged 36 cells. We imposed quality control criteria for the later analysis, specifically only cells with at least 30 sister pairs with both tracked for 75% of movie were retained; 31 cells satisfied this criteria giving 1281 tracked sister pairs. On average, we obtained 40 sister pairs per cell (quartiles Q1=38.5, Q3=43), where both sisters were tracked for at least 200 seconds (100 frames).

As is typical for mammalian cells, (*e.g.* HeLa cells [7,17]), KTs form a metaphase plate (Fig 1A) and undergo sawtooth oscillations perpendicular to the metaphase plate (Fig 1E) before separating in anaphase when KTs segregate towards their respective spindle poles (Fig 1C). RPE1 cells oscillate in metaphase with a period of 84s and have a median inter-sister distance (*i.e.,*Kinetochore-Kinetochore (KK) distance) of 1.04 microns during metaphase, averaged over cells and time, Fig 2A, 2C. The associated breather oscillations of the KK distance have double the frequency as expected, with a period of 44s (Fig G in S1 Appendix). By aligning a cell's KT trajectories to the median anaphase onset time of a cell (see Methods), we can quantify changes over time as anaphase is approached, *i.e.,*map metaphase maturation.

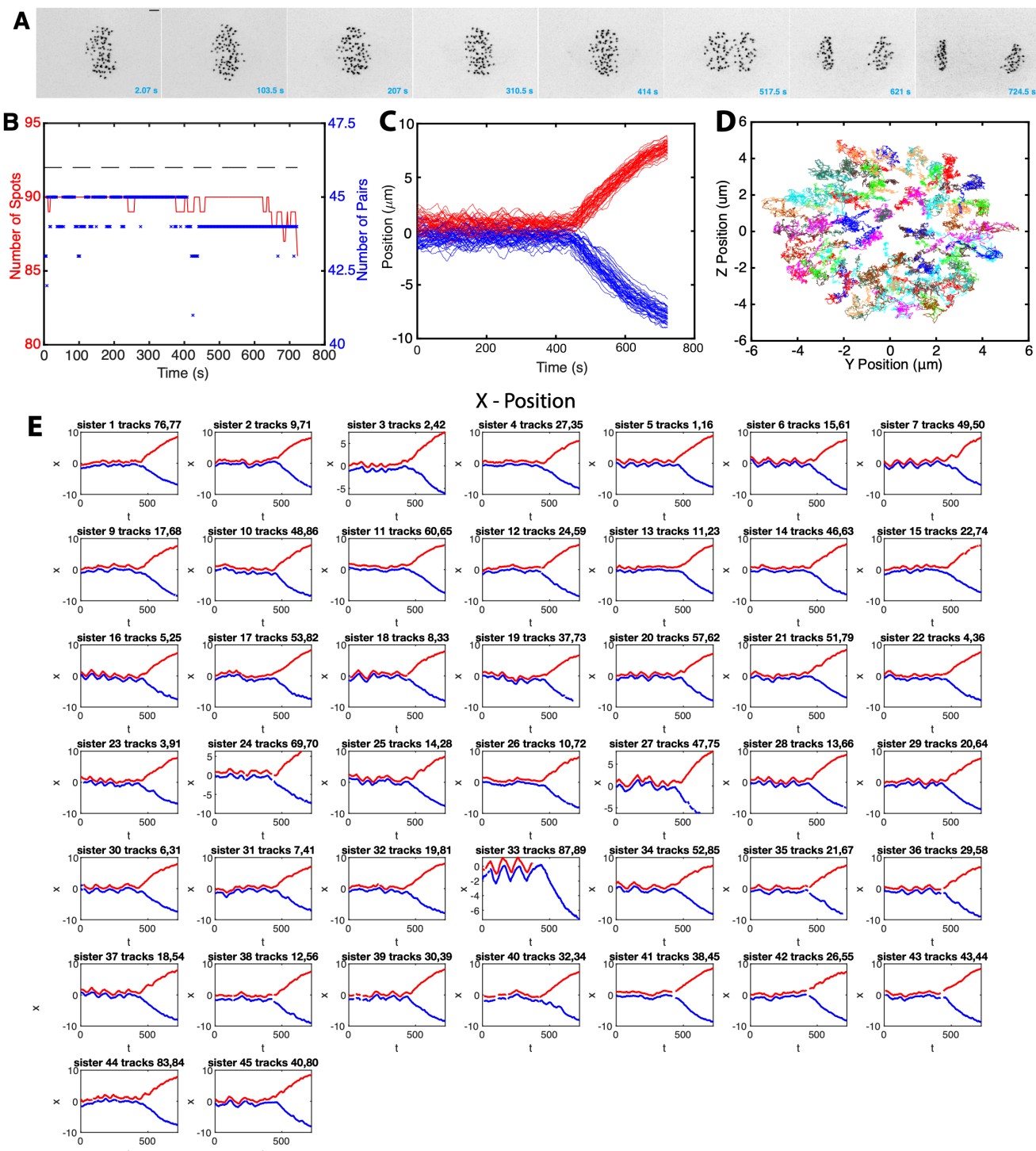

**Fig 1**. **Near-complete tracking of kinetochores through metaphase and anaphase in a human RPE1 cell. A** Sequence of *z*-projected LLSM images through metaphase and the metaphase-anaphase transition (movie duration 724.5 s (2 s/frame)). Scale bar 2 microns. **B** The number of KTs (spots) tracked through time (red). The number of KT pair trajectories at each time point (blue; only trajectories with both KTs tracked for at least 80% of the movie are shown). The dashed grey line indicates 92 kinetochores, the number in RPE1 cells. **C** Track overlay time course showing normal displacement of KT positions from the MPP; red and blue tracks demarcate KTs that descend to respective daughter cells. **D** *yz* overlay of tracks in metaphase, viewed from above the metaphase plate. **E** Individual KT pair tracks over time, red and blue show tracks of sister KTs. Data in A-E for one randomly chosen cell.

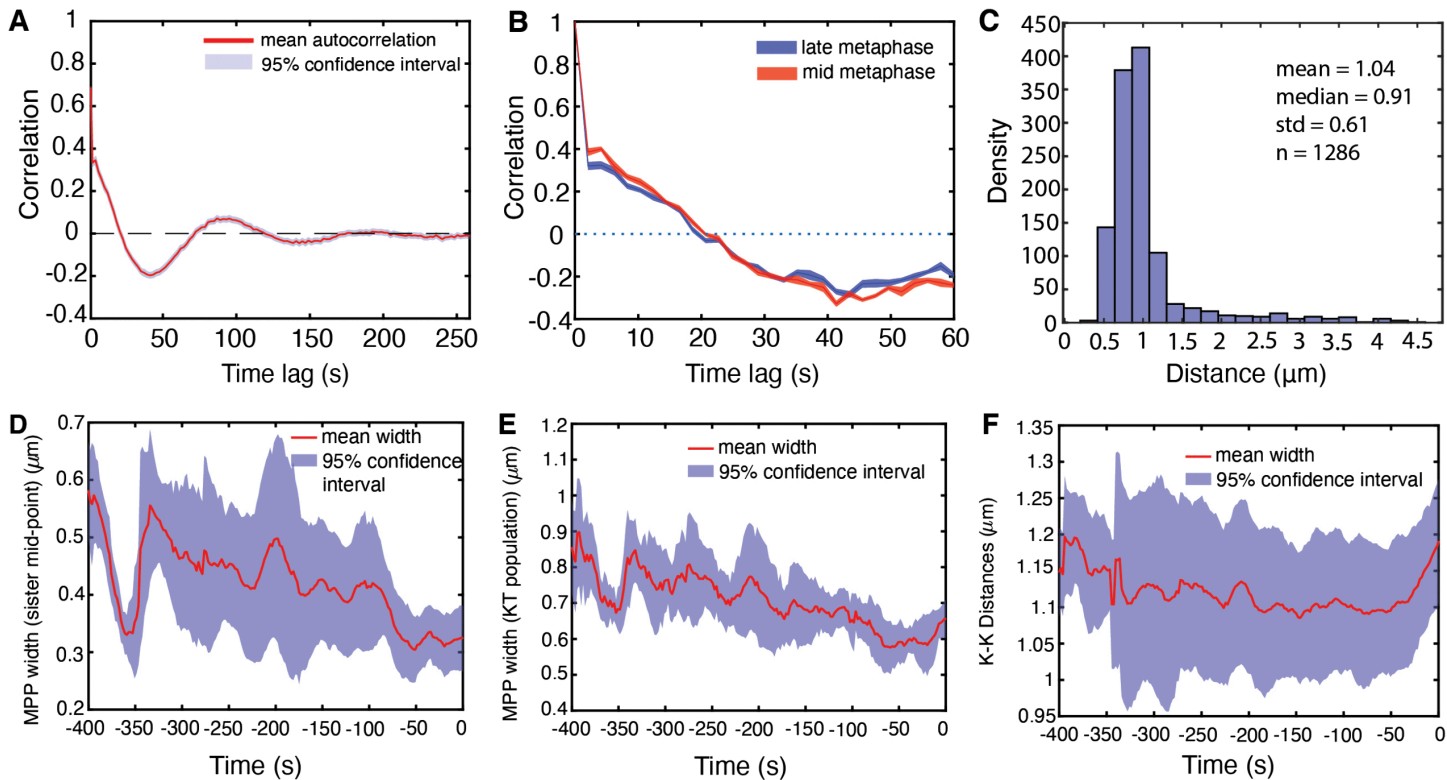

**Fig 2**. **Intrametaphase maturation dynamics. A** Autocorrelation function (ACF) of sister pair midpoints. **B** ACF of metaphase oscillations for late (red) and mid (blue) metaphase. **C** Inter-sister kinetochore (KK) distance pooled over all KT pairs and time. **D-F** Mean (red) and standard deviation (blue) of **D** MPP width as measured by paired sister mid-point width (smallest eigenvalue of the covariance matrix of kinetochore mid-points). **E** Metaphase plate (MPP) width as measured by the covariance matrix of the KT population (smallest eigenvalue), **F** KK distance against time to anaphase. Data are based on 31 cells having at least 30 sisters both tracked for 75% of movie.

We confirmed that the MPP becomes thinner over time, as measured by both the compaction of the sister mid-points (as used in [7]) (Fig 2D) and the full KT complement (Fig 2E). The sister KK distance reduces slightly over metaphase; it begins to increase sharply as some kinetochore pairs initiate anaphase (Fig 2F). We confirmed these changes by splitting metaphase into mid-metaphase (330-230 s before anaphase) and late metaphase (130-30s before anaphase, avoiding dynamics immediately prior to anaphase), showing that both the KK distance and the MPP width significantly decrease, ($p_{MW} < 10^{-3}$) (Fig B in S1 Appendix). However, the average oscillation period and strength is invariant mid to late metaphase (Fig 2B). Thus, the MPP width primarily decreases because sister pair oscillations centralise to the plate, with a smaller contribution from a reduction in oscillation amplitude (reduced average KK distance).

## 2.2 Modelling metaphase kinetochore dynamics

Here we outline the force balance model of [17] and its biological context, defining extensions of this model later in the text. We refer to the model of [17] as the *vanilla* model. The vanilla model is a submodel of all extensions, *i.e.,* it is reproduced by imposing appropriate parameter constraints. There are 4 forces acting on chromosomes (Fig 3A): K-fibers push or pull the chromosomes depending on their polymerisation state, pulling being the substantially stronger force, [17], the centromeric spring that connects the sister chromatids can be stretched or compressed thus generating an intersister force, centralising forces push the chromatids/KTs towards the cell mid-plane, and drag forces damp movements.

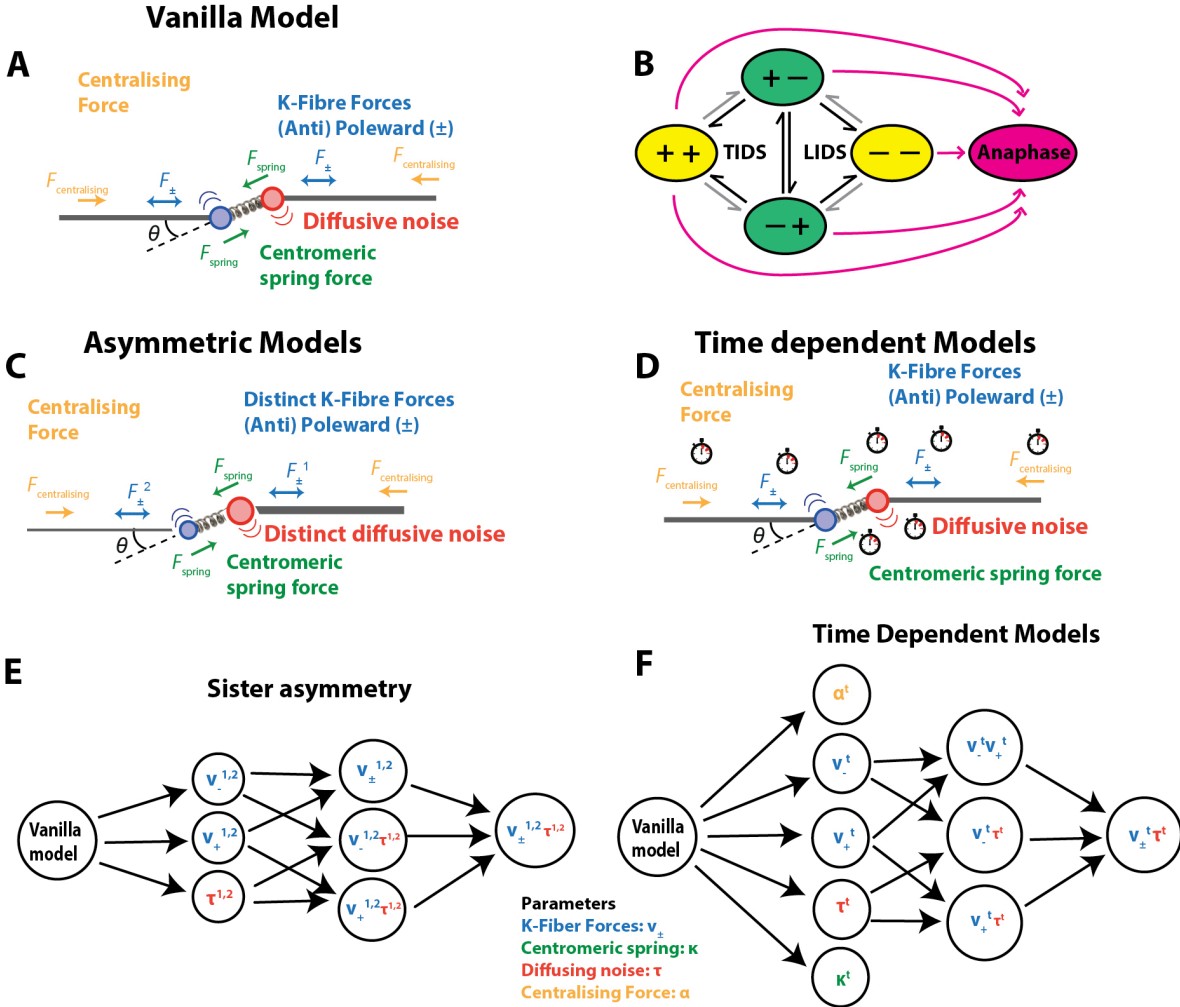

**Fig 3. Biophysical model schematics for KT dynamics in metaphase and transition to anaphase. A** Vanilla metaphase model showing the forces on a KT pair: K-fiber forces (either pulling (poleward, −) or pushing (anti-poleward, +)), spring force (green), centralising force (orange) and diffusive noise (arcs). Drag force not shown. **B** Schematic of hidden states comprising sister pair K-fiber states (+ pushing, − pulling). Transitions between states are either initiated by the leading sister (Leading sister Induced Directional Switch, LIDS) or the trailing sister (Trailing sister Induced Directional Switch, TIDS); a joint switch (both sisters switching direction in the same frame) is rare. Saw tooth oscillations arise because the coherent states (+−, −+) are of longer duration. Transitions in black depict the first sister switching during a directional switching event, gray the second. In models with anaphase, anaphase onset is an irreversible transition (pink), *i.e.,*anaphase is a absorbing state. **C** Schematic showing models with KT sister asymmetry in the K-fiber forces ($v_+$ or $v_-$) and/or in the diffusive noise ($\tau$). **D** Schematic showing models with time dependence in the biophysical parameters. **E** Nested model graph of the interrelationship between the asymmetric models. **F** Nested model graph of the interrelationship between the time dependent models. Colours refer to the biophysical parameters as in **A**, see key.

Following [17], we model the dynamics of the perpendicular distances of the two sister KTs from the MPP. Using force balance the dynamics are formulated as a pair of stochastic differential equations, but since time measurements are equi-spaced, with time interval $\Delta t$, we can integrate over the $\Delta t$ (assumed small). This gives the following discrete time model

for paired sister dynamics, (the *vanilla* model),

$$\left(X_{t+\Delta t}^1 - X_t^1\right) = \Delta t\left(-v_{\sigma_t^1} - \kappa\left(X_t^1 - X_t^2 - L\cos\theta_t\right) - \alpha X_t^1\right) + \sqrt{\Delta t}N\left(0, \tau^{-1}\right),$$
$$\left(X_{t+\Delta t}^2 - X_t^2\right) = \Delta t\left(+v_{\sigma_t^2} - \kappa\left(X_t^2 - X_t^1 + L\cos\theta_t\right) - \alpha X_t^2\right) + \sqrt{\Delta t}N\left(0, \tau^{-1}\right),$$

$$(1)$$

where $X_t^k$ is the perpendicular distance of sister $k$ from the MPP. Here $\sigma_t^k$ is the (polymerisation) state of the attached K-fiber to sister $k$ *i.e.*, either pushing ($+$) or pulling ($-$), with switching between states following a Markov model discussed below. The $v_{\sigma_t^k}$ term (taking values $v_+$ or $v_-$) is the K-fiber force acting on sister $k$ at time $t$. It is a composite force with two components, including the force generated from MT plus end (de)polymerisation dynamics within the K-fiber and the K-fibre retrograde flux. Specifically, we have the combined force $v_\pm = v_\pm^{tip} - v_{flux}$ where $v_{flux}$ is the K-fibre retrograde flux, and $v_\pm^{tip}$ is the tip (plus end) polymerisation, respectively depolymerisation force, (note $v_- \leq 0$, $v_+ \geq 0$). The term $\kappa\left(X_t^1 - X_t^2 - L\cos\theta_t\right)$ is a Hookean spring force modelling the centromeric chromatin spring (assumed linear) that connects sister KTs, spring constant $\kappa$, natural length $L$, and $\theta_t$ is the angle between the normal to the MPP and the vector connecting the sister pair at time $t$ (thereby projecting the spring force perpendicular to the MPP) (Fig 3A). The $\alpha X_t^1$ term models the centralising forces, a composite of the polar ejection force (PEF), (K-fiber) length dependent flux [16] and HURP dependent control of plus end dynamics (also length dependent, [45]), which all produce forces that push KTs/chromatids towards the cell mid-plane. We use a linear model for the centralising forces, specifically the leading order of a Taylor expansion in displacement, the next order being cubic by symmetry. Thus, the force is linear in the (perpendicular) displacement from the metaphase plate. Since the displacement is small relative to the length of the spindle and centralising forces are weaker than the K-fiber forces, [17], this will be a good approximation. We assume diffusive noise, giving the Gaussian noise distribution $N\left(0, \Delta t \tau^{-1}\right)$ in Eq (1), mean 0, variance parametrised by the precision $\tau$ (inverse of variance). All forces are divided by the unknown drag coefficient as in [17]; the effect of this is that all terms in the parentheses in Eq (1) have dimensions of speed, and the force parameters $\alpha$ and $\kappa$ have units $\left[s^{-1}\right]$. The drag force is also likely a composite, with contributions from viscous drag from the cytosol and drag from the MT matrix.

A sister pair at time $t$ can be in any of 4 possible polymerisation states, $\sigma_t \in \{++, +-, -+, --\}$ ($+$ polymerisation/pushing, $-$ depolymerisation/pulling), the state $\sigma_t$ evolving as a discrete time Markov chain with transition matrix $P$,

$$P = \begin{bmatrix} p_{incoh}^2 & p_{incoh}q_{icoh} & p_{incoh}q_{icoh} & q_{incoh}^2 \\ p_{coh}q_{coh} & p_{coh}^2 & q_{incoh}^2 & p_{coh}q_{coh} \\ p_{coh}q_{coh} & q_{coh}^2 & p_{incoh}^2 & p_{coh}q_{coh} \\ q_{icoh}^2 & p_{incoh}q_{coh} & p_{incoh}q_{icoh} & p_{incoh}^2 \end{bmatrix},$$

$$(2)$$

where $p_{coh}$ and $p_{incoh}$ are the probabilities of a KT remaining in a coherent state (sisters move in the same direction, states $+-, -+$), respectively incoherent state (sisters move in opposite direction, states $++, --$) over a time interval $\Delta t$. Here $q_{coh} = 1 - p_{coh}$ and $q_{incoh} = 1 - p_{incoh}$.

Simulations of this biophysical model, Eqs (1), (2), produce trajectories with quasi-periodic oscillations qualitatively similar to observed data, [17]; saw-tooth like oscillations occur when the coherent state mean lifetime is larger than the incoherent state lifetime. We use a Bayesian approach to fit the dynamic models to each experimental sister pair trajectory, see Methods. Bayesian approaches jointly infer all the model parameters and propagate uncertainty. We imposed additional quality control for model inference, removing cells with less than 120 frames and trajectories with over 20% missing data, Section C1 in S1 Appendix. This gave 26 cells for the model analysis.

## 2.3 Biophysical characteristics of quasi-periodic oscillations in diploid non-transformed human RPE1 cells

We inferred the biophysical parameters for RPE1 cells on the vanilla model, (Eqs (1), (2)), using Bayesian inference methods that infer all model parameters concurrently, see Methods and Section C in S1 Appendix. We analysed (after data quality control) 798 KT pairs from 26 cells (average of 31.9 sister pairs/cell). An example of model inference is shown in Fig 4 for a single sister pair trajectory, demonstrating that all parameters are inferrable with reasonable posterior variances. Recall that the natural length $L$ is not identifiable so the posterior is nearly identical to the prior as expected, [17]. Fitting the model to the trajectory provides an automated state annotation, *i.e.,*the hidden K-fiber pulling/pushing state is inferred at each time point (Fig 4B) thereby allowing switching events of the KT sister pair state to be inferred (Fig 4C). Metaphase quasi-periodic oscillations require that both KTs of a sister pair change direction at a directional switch, [17]. There are two switching choreographies: a switching event initiated by the leading sister (lead induced directional switch, or LIDS) and switching initiated by the trailing sister (trail induced directional switch, or TIDS) (Fig 3B); coincident switching where directional switching is faster than the time resolution is also observed, but is in the minority. In this trajectory directional switching (between coherent states $+-$ to $-+$, or $-+$ to $+-$), occurs through LIDS events, *i.e.,*the intermediate state is $++$, compressing the centromeric spring. Both choreographies are in fact observed with a LIDS/TIDS ratio of 3.2, specifically 68.5% LIDS, and 21.1% TIDS and 10.3% joint switches (26 cells, 798 KT pairs). This is similar to that previously reported for HeLA cells, with a LIDS/TIDS of 3.8, [17].

Inferred biophysical parameters for RPE1 cells are given in Table 1. We note that these are similar to previously inferred parameters in HeLA cells [17].

## 2.4 Sister kinetochores exhibit substantial sister asymmetry in kinetochore forces

The two spindle poles are in fact not equal, respective spindle poles having a young and old centrosome. Centrosome age affects the stability of MTs from the associated spindle pole which imparts a half-spindle bias, both in positioning of unaligned KTs and missegregation, [46]. Thus, assuming KT sister dynamics is symmetric is not justified. The vanilla model assumes sister KTs and K-fibers are biophysically identical (symmetric) and thus have the same pulling and pushing forces. Here, we relax the assumption of sister symmetry, *i.e.,*sister $k = 1, 2$ can have distinct (unloaded) forces $v_\pm^k$ and different levels of diffusive noise, specifically different precisions $\tau^k$ in Eq (1), (Fig 3C). Clearly spring constant $\kappa$ is common. We do not consider asymmetry in $\alpha$ here, but note that asymmetry in $\alpha$ could arise from asymmetry in the PEF due to an asymmetric spindle, or asymmetry in the centralising flux. We confirm later that there is no half-cell bias in $v_\pm$ supporting the assumption that the spindle poles are of equal strength, *i.e.,*the centralising forces should be symmetric.

The most general asymmetric model we consider is given by, (generalising model Eq (1))

$$X_{t+\Delta t}^1 = X_t^1 + \Delta t \left( -v_{\sigma_t^1}^1 - \kappa \left( X_t^1 - X_t^2 - L \cos \theta_t \right) - \alpha X_t^1 \right) + \sqrt{\Delta t} N(0, (\tau^1)^{-1}),$$

$$X_{t+\Delta t}^2 = X_t^2 + \Delta t \left( +v_{\sigma_t^2}^2 - \kappa \left( X_t^2 - X_t^1 + L \cos \theta_t \right) - \alpha X_t^2 \right) + \sqrt{\Delta t} N(0, (\tau^2)^{-1}), \tag{3}$$

where the hidden states $\sigma_t^k \in \{+, -\}$ have the same dynamics as above, Eq (2). This model allows sisters to differ in 3 variables $(v_\pm, \tau)$; we also consider the reduced models with only 1 or 2 parameters being different, giving 7 asymmetric models in addition to the vanilla (symmetric) model (Eq (1)). The relationship (nesting) of these models is shown in Fig 3E. A more complex model is deemed to be supported from the data if the Bayes factor $B > 3.2$, see Methods. We report results of the following 5 models:

- $M_0$ no asymmetries, *i.e.,*the symmetric vanilla model (Eq (1)),
- $M_{v_-}^a$ asymmetry only on $v_-$,
- $M_{v+}^a$ asymmetry only on $v_+$,

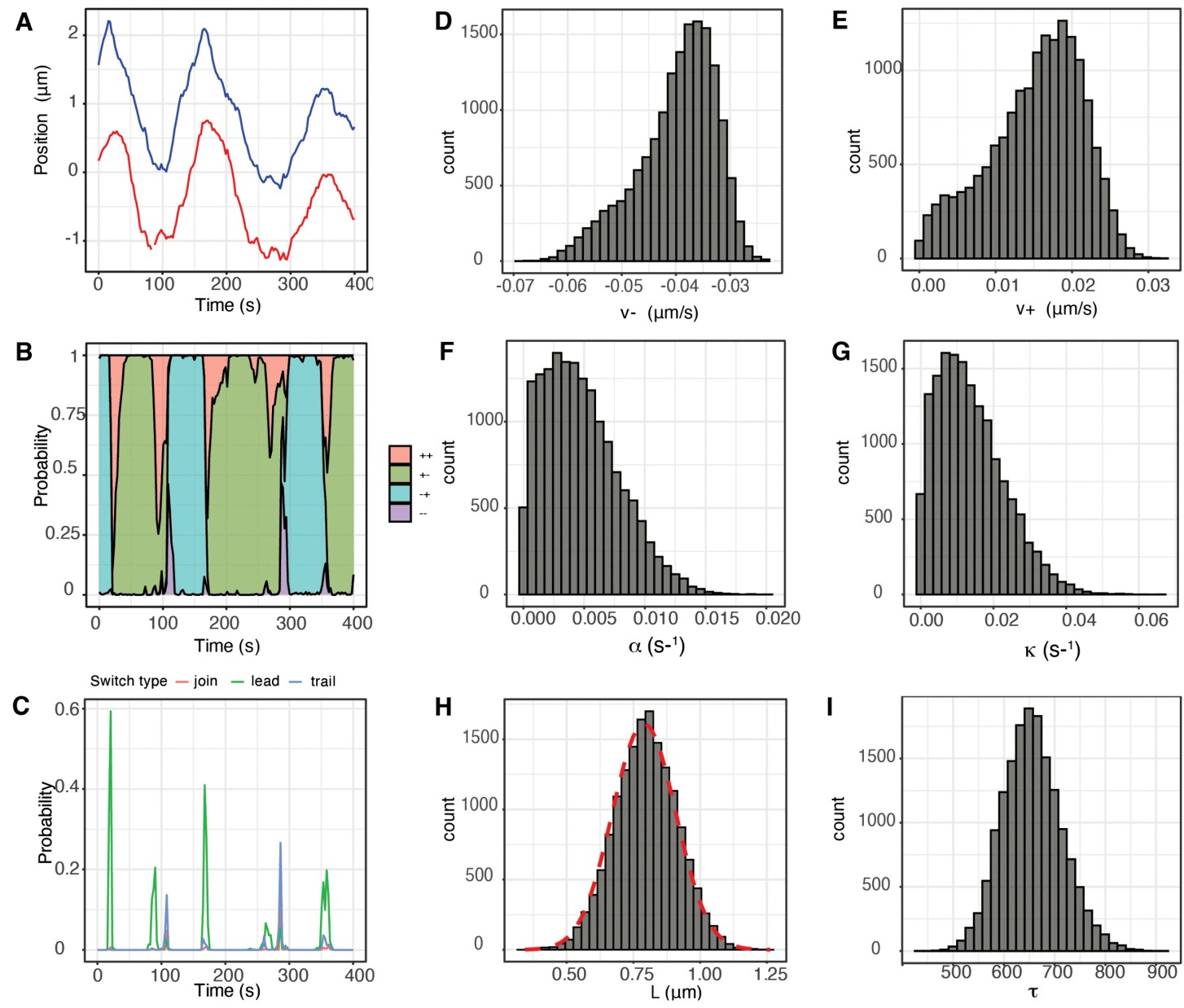

**Fig 4. Single KT pair trajectory inference showing model parameter estimates and trajectory annotation. A** KT distance transverse to the MPP for the 2 sisters (red, blue). (LLSM imaging of Ndc80-eGFP endogenously marked KTs). Additional data for this KT pair is given in Fig F of S1 Appendix. **B** Trajectory annotation for the hidden K-fiber polymerisation states; probability $P(\sigma_t|x_{1:T};\theta)$ for being in each state at each time point as sampled via the backward algorithm (see Section C3 of S1 Appendix). **C** Probability of a directional switch initiated by the leading kinetochore (green), trailing kinetochore (grey), or a joint switch (red). Switching probability is determined using the sampled hidden states and computed as the proportion of MCMC samples matching a particular pattern of states (e.g. LIDS $[-+,-+,++,+-]$) associated with a given switch type. **D-I** Marginal posterior distribution of the stated biophysical parameter for the trajectory data in **A**. In **H** the informative prior density for natural length $L$ is plotted (red). All other parameters have uninformative priors.

- $M_{v\pm}^a$ asymmetries on $v_-$ and $v_+$,
- $M_{v\pm\tau}^a$ asymmetries on $v_-$, $v_+$ and $\tau$,

**Table 1**. **Biophysical parameters of RPE1 cells inferred on the vanilla model, Eqs (1), (2).** The median of posterior medians is shown in the second column with interquartile range (IQR) in brackets. The third column shows the median of the Posterior Interquartile Range (PIR), typically less than the IQR. Based on 26 cells and 798 KT pairs.

| Parameter | Median [IQR] | Median PIR |
|---|---|---|
| $\tau$ | 673.7 [241.4] | 95.04 |
| $\kappa$ (s$^{-1}$) | 0.022 [0.018] | 0.018 |
| $\alpha$ (s$^{-1}$) | 0.009 [0.008] | 0.005 |
| $v_-$ ($\mu$m/s) | −0.041 [0.021] | 0.008 |
| $v_+$ ($\mu$m/s) | 0.010 [0.010] | 0.007 |
| $L$ ($\mu$m) | 0.808 [0.033] | 0.158 |
| $p_{incoh}$ | 0.760 [0.216] | 0.155 |
| $p_{coh}$ | 0.949 [0.027] | 0.026 |

with the superscript $a$ indicating that the model is asymmetric and the subscript specifying which parameter(s) are asymmetric. The other models were rarely preferred, so not discussed further. The model likelihoods for these 5 models are given in Section C in S1 Appendix.

We fitted the 4 asymmetric sister models to 798 sister pairs, across 26 cells. In RPE1 cells, 24.4% of sister kinetochores have significant asymmetry, with most, (45.1% of asymmetric pairs) preferring asymmetry in both $v_-$ and $v_+$ whilst 88.7% have significant asymmetry in $v_-$ (preferring any of the models $M_{v_-}^{\alpha}$, $M_{v_- v_+}^{\alpha}$, $M_{v_- v_+ \tau}^{\alpha}$) (Fig 5A and Table A in S1 Appendix). However, (significantly) asymmetric sisters only show small differences in biophysical parameters relative to symmetric sisters, Table 2; asymmetric and symmetric trajectories in fact appear similar (Fig 5C). Asymmetric sisters have a significantly larger (9.3%) average pulling force (averaged over the two sisters) than symmetric sisters ($p_{MW} < 10^{-7}$, one-sided), whilst the centromeric spring constant, mid-plane centralising forces and natural length are 15.9 % ($p_{MW} =$ 0.01), 34.3% ($p_{MW} < 10^{-7}$) and 2.6% ($p_{MW} < 10^{-6}$) larger. Note that details and abbreviations on the statistical tests used in throughout our analysis can be found in Table O of S1 Appendix. Since it is easier to detect differences when $|v_-|$ is larger, the asymmetric group would be expected to have a larger average $v_-$. In all cells there is a fraction of (significantly) asymmetric sisters, see Fig E(A,B) in S1 Appendix, while, there is also no half-spindle bias in the (significantly) asymmetric KTs, $p_{Binom} = 0.88$, *i.e.,* cells exhibit equal numbers of stronger KTs to the right, respectively left, half-spindle (Fig E(C,D) in S1 Appendix). Thus, kinetochore asymmetry is not due to a spindle asymmetry that could arise because of the spindle poles having a younger, respectively, older centrosome, [46].

We examined if pushing and pulling forces are independent for individual KTs. There is a significant anti-correlation between pushing ($v_+$) and pulling ($|v_-|$) forces in the asymmetric population (Fig 5D), specifically K-fibers that are stronger at pulling are typically weaker at pushing ($p_{Binom} < 10^{-8}$, rejecting the null hypothesis that the strongest pulling sister ($v_-$) had an equal probability of being either the stronger or weaker pushing sister ($v_+$)). Thus, pairs where a sister is both a stronger puller and pusher (respectively weaker puller and weaker pusher) are in a minority. The difference in sister forces, $\Delta v_{\pm} = v_{\pm}^1 - v_{\pm}^2$, shows this trend on all KT pairs (Fig 5D) suggesting asymmetry may be a natural consequence of KT and K-fiber variation. Sister differences were in fact not significantly different to differences between any 2 randomly chosen KTs, (the force difference distributions were identical, $p_{MW} = 0.986$, $p_{MW} = 0.621$, for $\Delta v_- = v_-^1 - v_-^2$ and $\Delta v_+ = v_+^1 - v_+^2$, respectively, and $p_{KS} = 0.056$ and $p_{KS} = 0.141$ for $\Delta v_-$, $\Delta v_+$, respectively) (Fig 5E). However, there is variability in the biophysical parameters due to location across the MPP that increases the variance of the population, *i.e.,* the population is spatially heterogeneous, Section 2.6. To rule out that the lack of significance is due to this variance contribution we applied the same tests on 5 groups of KT pairs grouped by their average radial location in the MPP, as in Section 2.6. The lowest p-value for the tests over the 5 groups was $p_{MW/KS} = 0.385$, indicating that sister KT variation is similar to that between 2 randomly chosen KTs with similar locations in the MPP. This is further explored in Fig AG(A) in S1 Appendix.

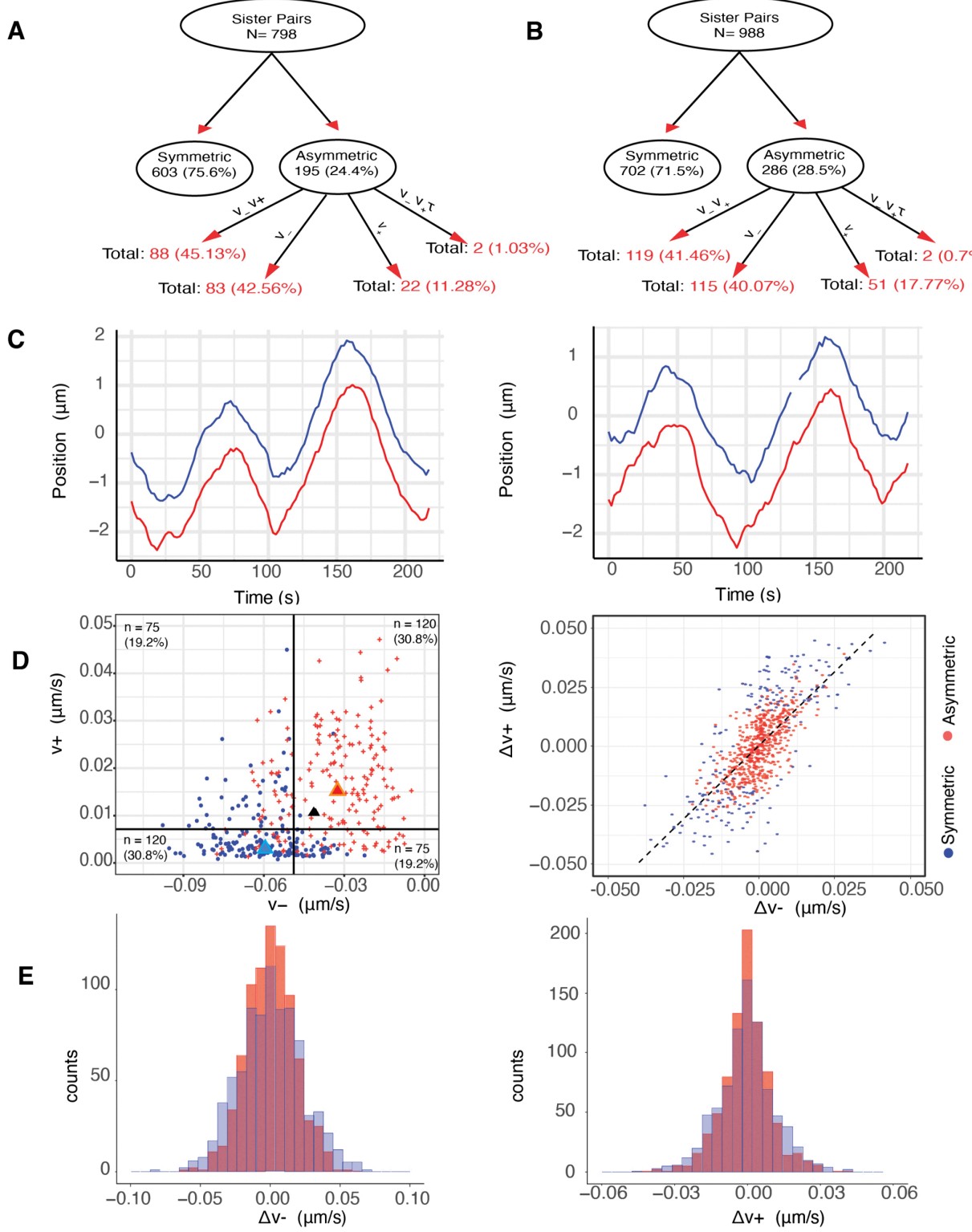

**Fig 5. Asymmetry in sister kinetochores is a result of natural variation (noise). A/B** Graphical representation of the asymmetry model preference network with decomposition shown as percentage of parent node in (A) DMSO and (B) nocodazole washout treated cells. **C** A typical symmetric (left) and asymmetric (right) trajectory, (DMSO). **D** (Left) Correlation of inferred $v_-$ and $v_+$ of asymmetric sister pairs (inferred on the asymmetric $v_{\pm}$ model, $M_{v_{\pm}}^a$). For each pair, the stronger (blue) and weaker (red) pulling sister are shown. Population medians are marked with triangle of the same colour, while the black triangle shows the corresponding population median of the symmetric pairs. The number and percentages of KTs in each quadrant is

shown (relative to the median of the plotted data). (Right) Scatter plot of sister difference in pulling force $\Delta v_- = v_-^1 - v_-^2$ versus pushing force $\Delta v_+ = v_+^1 - v_+^2$, with symmetric and asymmetric sister pairs indicated in red and blue respectively, (inferred on the $v_\pm$ asymmetric model). Asymmetric pairs do not present any distinctive pattern. **E** Histograms of the sister difference distribution, *i.e.,*$\Delta v_-$ (left) and of $\Delta v_+$ (right), when sisters are paired (red) and pairing is randomised (blue), (randomisation takes place within a MPP radial group to account for variation with respect to the metaphase plate radial position, see Fig 7).

**Table 2**. **Biophysical parameters of symmetric and (significantly) asymmetric sister pairs.** Sister 1 is the sister with the greater pulling force in an asymmetric pair (i.e. $|v_-^1| \geq |v_-^2|$). The second and fourth columns report the medians of posterior medians for symmetric and asymmetric sisters respectively; the population interquartile range (IQR) is shown in brackets. The average parameter confidence per trajectory is shown in the third and fifth columns, specifically the median of Posterior Interquantile Ranges (PIR) for symmetric and asymmetric sisters, respectively. Based on Mann-Whitney and Kolmogorov-Smirnov tests, the asymmetric posterior distributions of $\kappa$, $\alpha$, $v_-$ and $L$ are significantly different from the symmetric posterior distributions, with effect sizes (percentage difference of the medians) of 15.9%, 34.3%, 9.5%, and 2.6%, respectively. Inference on asymmetric sisters is based on $M_{v_\pm}^\alpha$.

| Parameter | Symmetric | | Asymmetric | |
| --- | --- | --- | --- | --- |
| | Median | PIR | Median | PIR |
| $\tau$ | 661.8 [265.4] | 95.055 | 700 [311.9] | 96.90 |
| $\kappa$ | 0.020 [0.018] | 0.017 | 0.023 [0.022] | 0.017 |
| $\alpha$ | 0.010 [0.008] | 0.005 | 0.013 [0.010] | 0.005 |
| $v_-^1$ | −0.042 [0.021] | 0.008 | −0.060 [0.020] | 0.009 |
| $v_-^2$ | −0.042 [0.021] | 0.009 | −0.032 [0.026] | 0.009 |
| $v_+^1$ | 0.010 [0.010] | 0.007 | 0.004 [0.005] | 0.007 |
| $v_+^2$ | 0.010 [0.010] | 0.007 | 0.015 [0.016] | 0.007 |
| $L$ | 0.806 [0.033] | 0.156 | 0.839 [0.067] | 0.155 |
| $p_{incoh}$ | 0.734 [0.216] | 0.173 | 0.766 [0.215] | 0.133 |
| $p_{coh}$ | 0.950 [0.027] | 0.026 | 0.952 [0.024] | 0.024 |

We conclude that K-fiber forces are intrinsically variable and therefore sister differences (asymmetry) is a continuum. The significantly asymmetric sisters will therefore be those with the largest differences. Notably, the mechanism underlying this KT/K-fiber stochasticity introduces an inherent anti-correlation between pulling and pushing forces.

## 2.5 Symmetric and asymmetric sisters are organised transversely to the MPP

Our analysis reveals that K-fibers are intrinsically noisy, which results in an asymmetry in sister KTs in $v_\pm$ and in the diffusive noise $\tau$ (Fig 5). Sister asymmetry in pulling forces should localise the sister pair off centre of the MPP plate, the K-fiber pulling force being the dominant force acting on the chromosome, [17]. Specifically, a sister pair with a substantially stronger pulling sister will be pulled on average towards its pole, experiencing on average a higher centralising force away from that pole, the pair thus achieving a stable average off-centre position towards the stronger sister's pole. This is in fact observed, significantly asymmetric sister pairs are localised on average towards the stronger sister's pole relative to symmetric sisters, (Fig 6A). To ascertain if this repositioning is retained into anaphase, we back-tracked the KTs that descend to respective poles tracking the cluster back through metaphase. Within each poleward destined cluster, we partition the KTs into 3 groups, firstly those that are part of a symmetric pair, secondly the stronger pulling sister of an asymmetric pair and finally the weaker pulling sister of an asymmetric pair. The stronger pulling sisters are positioned towards the pole in metaphase (Fig 6C), as expected since the sister centre is offset towards the stronger sister's pole, an offset that is significant relative to the KTs from symmetric pairs, $p_{MW/KS} < 10^{-16}$. This offset is in fact retained through into anaphase (Fig 6E). Similarly, the KTs that have the weaker pulling force of an asymmetric pair are positioned away from the pole to which they descend, $p_{MW/KS} < 10^{-14}$ (Fig 6C, 6E). An analysis of the KT position ranking within its cluster (ranked mean position towards the pole) also demonstrates the relative positioning of the weaker and stronger sisters of an asymmetric pair compared to kinetochores of a symmetric pair (Fig Q in S1 Appendix). The relative positioning is in fact fairly stable through metaphase with a small number of KTs changing position – what is surprising is that asymmetric KTs tend to be at

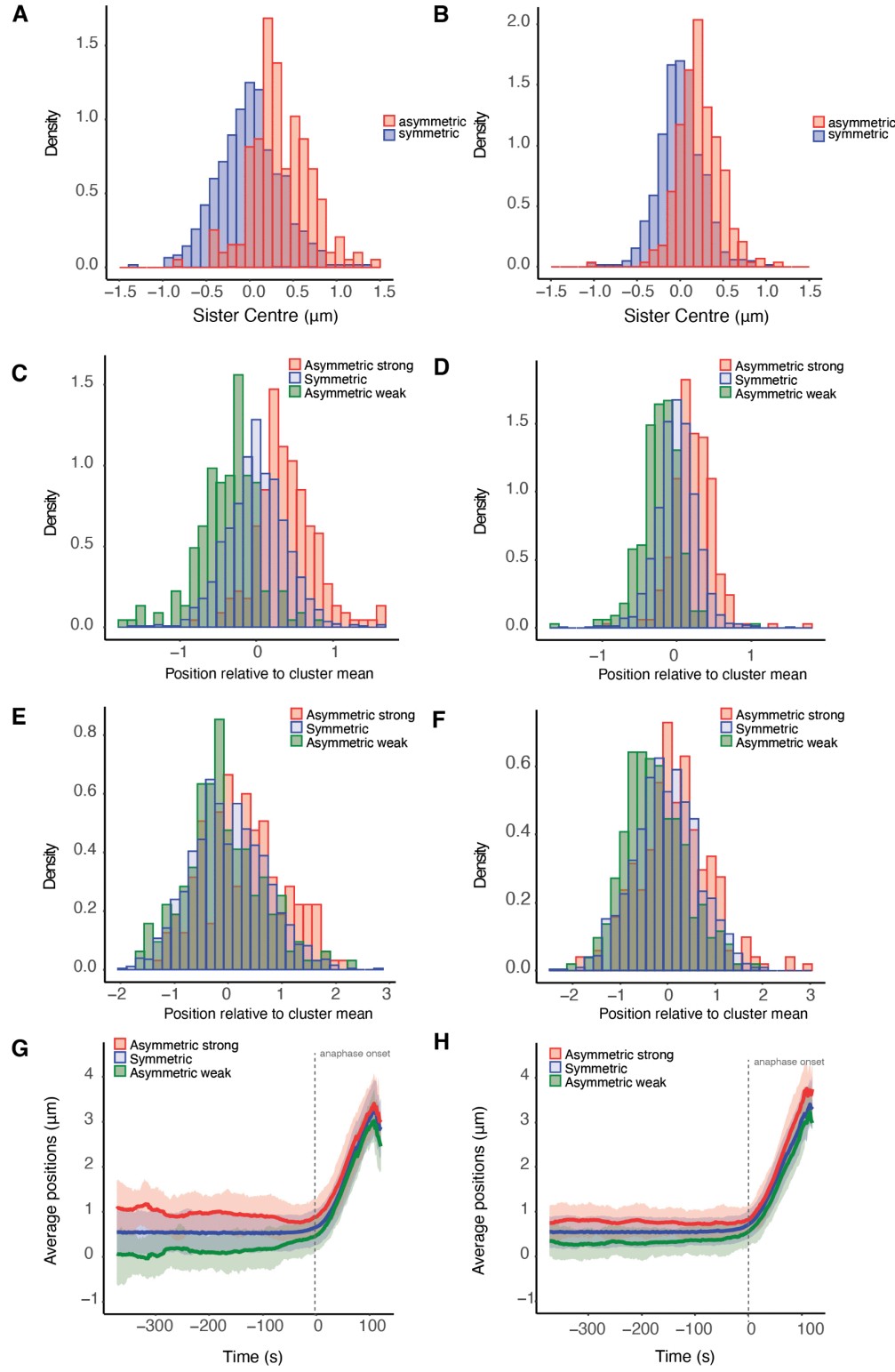

**Fig 6. Symmetric and asymmetric kinetochores are organised transverse to the MPP.** Transverse organisation of the KTs across the MPP by K-fiber pulling strength grouped into stronger, weaker KT of an asymmetric pair, and KTs in a symmetric pair, for: left (**A,C,E,G**) DMSO (195 asymmetric and 603 symmetric sister pairs), right (**B,D,F,H**) nocodazole washout treated cells (286 asymmetric and 702 symmetric sister pairs). **A/B** KT pair average

position (over sisters and time) normal to the MPP. For asymmetric KT pairs, positive distance corresponds to movement towards the stronger pulling sisters' pole. **C/D** Mean displacement of KTs relative to the mean position of their relevant (anaphase) cluster during metaphase. Stronger (weaker) asymmetric asymmetric sisters are positioned closer to (away from) their pole. **E/F** Mean displacement of KTs relative to the mean position of their cluster in anaphase. The median position of asymmetric stronger pulling sisters is significantly greater than the median position of symmetric sisters which, in turn, is significantly greater than the median position of asymmetric weaker sisters for both DMSO and nocodazole washout treated cells, ($p_{MW} < 10^{-5}$, $p_{MW} < 10^{-10}$). **G/H** Time evolution of the average positions in time. Shaded envelope shows interquartile range (IQR) for each group. Time is measured relative to anaphase at $t = 0$, *i.e.* the median anaphase onset time over all KT pairs in a cell.

the extremes, the majority in their correct position (rank), but a minority diametrically opposite (Fig Q(H,I) in S1 Appendix). Our results demonstrate that there is an ordering along the spindle axis (MPP normal) of KTs that is a consequence of unequal K-fiber pulling forces in metaphase, an ordering that is retained through anaphase (Fig 6G). Thus, stronger KTs of a sister pair are positioned in the front of the cluster, weaker KTs at the back in both metaphase and anaphase.

### 2.6 K-fiber parameters have substantial spatial variation across the metaphase plate

Previous reports suggest that KT dynamics varies with position in the MPP, in particular that (i) trajectories are more stochastic towards the periphery, [7,17] (ii) that the centralising force increases towards the periphery of the MPP, [17] and (iii) that sister pairs at the periphery exhibit higher swivel, [11]. To determine if there are trends in the biophysical parameters with distance $r$ from the centre of the metaphase plate in RPE1 cells, we partitioned the population of KTs with respect to position, as in [17], (Fig 7). There are substantial trends with $r$ in a number of parameters, specifically the pulling force $v_-$ ($p_{Corr/Kendall/Spearman} < 10^{-16}$, Fig 7C) and pushing force $v_+$ ($p_{Corr} < 10^{-7}$, $p_{Kendall/Spearman} = 0.002$, see Fig 7D), the spring constant $\kappa$ ($p_{Corr} < 10^{-8}$, $p_{Kendall/Spearman} < 10^{-13}$, Fig 7F) and precision $\tau$, ($p_{Corr/Kendall/Spearman} < 10^{-16}$, Fig 7H). Trends are also detectable in $p_{incoh}$ and $p_{coh}$ (Fig I in S1 Appendix), while these trends are also detectable in single cells, (Fig J and Table F in S1 Appendix). In contrast to previous reports (on HeLA cells), there was no trend in the centralising force, which was invariant with $r$ ($p_{Corr} < 0.48$, $p_{Kendall/Spearman} = 0.12$), see Table E in S1 Appendix for a summary of p-values. In the previous study on HeLa cells, [17], no trends in the other dynamic parameters were observed in contrast to those observed here in RPE1 cells. The coverage in that study was substantially lower than in our study, which may explain these differences; however, they could also reflect differences between RPE1 and HeLa cells. Finally, we address if changes in drag can explain these trends. Chromosome size is known to increase towards the periphery, [47, 48], which could increase drag forces towards the periphery, whilst drag may also be dependent on MT density, that may vary laterally across the spindle. A change in the drag coefficient with $r$ scales all the mechanical parameters as follows, generalising Eq (1):

$$\left(X_{t+\Delta t}^k - X_t^k\right) = \gamma(r)^{-1}\left(-v_{\sigma_t^k} - \kappa\left(X_t^1 - X_t^2 - L\cos\theta_t\right) - \alpha X_t^k\right)\Delta t + \sqrt{\Delta t}N(0, (\tau\gamma(r)^2)^{-1}). \tag{4}$$

Thus, $v_\pm, \kappa, \alpha, \tau^{-1/2}$ would scale identically with $r$. Note that allowing for Einstein's relation between diffusion and drag would give $\tau^{-1/3}$. The scalings are compared in Fig 7B. There is substantial variation in the relative reduction in the parameters (percentage difference from periphery to the middle of the MPP 38%, 25%, 9%, 53% and 2% of $v_-, v_+, \alpha, \kappa$ and $L$ respectively), whilst an opposite trend is seen for $\tau^{-1/2}$, which increases by 10%. This is consistent with the observation that trajectories become more stochastic towards the MPP periphery. This suggests that solely changing drag across the MPP is not consistent with the data. The relative changes in the parameters are also not consistent with changing both drag and chromosome size across the MPP; a change in chromosome size would affect both the drag and the PEF.

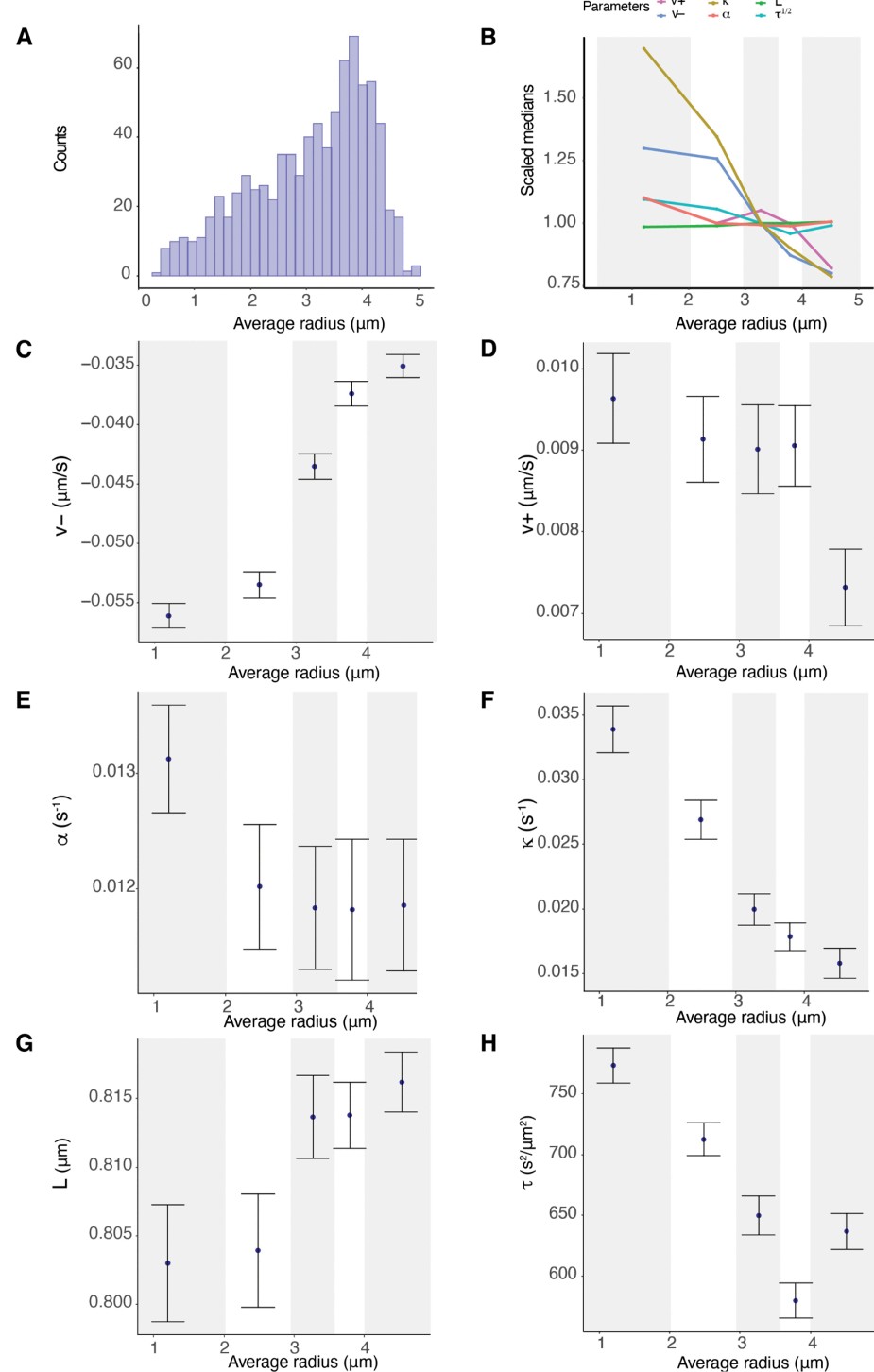

**Fig 7**. **K-fiber pulling strength and centromeric spring stiffness decreases towards the periphery of the metaphase plate. A** Radial location of KT pairs within the metaphase plate across 26 cells. **B-H** Summary statistics (medians of posterior medians and standard deviation) of indicated biophysical parameters partitioned into 5 radial groups (of the average KT pair radial position within the MPP). Groups contain the same number of observations. The five partitions are: $[0, 2.03]$, $(2.03, 2.95]$, $(2.95, 3.58]$, $(3.580, 4.01]$, $(4.01, 5.03]$. **B** Summary of spatial trends over all biophysical parameters. **C** Pulling forces $v_-$, **D** Pushing forces $v_+$, **E** Centralising force parameter $\alpha$, **F** Spring constant $\kappa$, **G** Natural length of the centromeric chromatin spring connecting the kinetochore sisters $L$, **H** precision $\tau$. Based on 798 KT pairs over 26 cells.

Biological variation is expected both within cells, resulting in differences in the mechanical properties of individual KTs and K-fibers, and between cells, with the between cell variation expected to be substantially larger. However, the variation we see with respect to location in the metaphase plate in our pooled cell samples indicates that within-cell variation is in fact substantial since it is evident even on data pooled over cells. To ascertain the contributions to the variance from spatial location and biological variation between cells we performed a 2-way ANOVA on the median estimates per sister pair for each parameter (Table 3), grouping with respect to the 5 radial partitions and the 26 cells. The interaction term (between radial location and cell) was always insignificant (in all parameters), so removed from the ANOVA analysis. In all parameters cell grouping was significant, whilst MPP location was significant in all except $\alpha$. However, the variance contributions from location and cell showed high variability. Location dominates for $v_-$ and $p_{incoh}$, indicating that K-fiber pulling forces and the duration of directional switching is heterogeneous with location (Table 3). All other parameters have similar (or smaller) variance contributions from location compared to between cells. Recall that we use a strong prior for the natural length $L$ because of identifiability issues [17]. The fact that there are significant differences between cells in $L$ possibly suggests true cell variability, although the size of effect is small. The higher between cell variance of $\alpha$ suggests that the centralising force varies substantially between cells, whilst K-fibre pulling force and the centromeric spring vary substantially within cells, consistent with the strong trends seen in Fig 7. The relative (total) standard deviation (sd/mean) shows that variation of $v_\pm, \alpha, \kappa$ is substantial (over 30%), which is clearly evident in the parameter estimates by cell (Fig H in S1 Appendix).

## 2.7 K-fiber mechanical parameters are time dependent and tune towards an anaphase ready state

The metaphase plate (MPP) undergoes time-dependent maturation in RPE1 cells, including a compaction of the width along the spindle axis as anaphase onset is approached, (Fig 2); with similar behaviour observed in HeLa cells [7]. We therefore investigated whether the biophysical parameters change in time, potentially explaining MPP maturation. We modified the vanilla model, Eq (1) to have time dependent parameters; we used an exponential time dependence as this preserves the parameter's sign whilst it is also a good approximation for weak linear time dependence. The full time dependent model has 5 time dependent parameters (Fig 3D, 3F) and is given by,

$$X^1_{t+\Delta t} = X^1_t - v_{\sigma^1_t,0}e^{v_{\sigma_t,1}\,t}\Delta t - \kappa_0 e^{\kappa_1 t}\Delta t(X^1_t - X^2_t - L\cos(\theta_t)) - \alpha_0 e^{\alpha_1 t}\Delta t X^1_t + \sqrt{\Delta t}N(0,\tau_0^{-1}e^{-\tau_1 t})$$
$$X^2_{t+\Delta t} = X^2_t + v_{\sigma^2_t,0}e^{v_{\sigma_t,1}\,t}\Delta t - \kappa_0 e^{\kappa_1 t}\Delta t(X^2_t - X^1_t + L\cos(\theta_t)) - \alpha_0 e^{\alpha_1 t}\Delta t X^2_t + \sqrt{\Delta t}N(0,\tau_0^{-1}e^{-\tau_1 t})$$

(5)

where we define for a time dependent parameter the constant term, with subscript 0, (value at the start of the movie), and rate of change parameter (on a log scale) with subscript 1. We don't analyse time dependence in the switching rates per frame ($1 - p_{coh}, 1 - p_{incoh}$), as there are too few switching events per trajectory, or for time dependence of the centromeric springs' natural length as that is determined from an accessory experiment (nocodazole treatment) because of identifiability issues. We consider a nested sequence of models (Fig 3F). We label models by the parameters that are time dependent, for example $M^t_{v_+}, M^t_{v_-}$ are the models with only $v_+$, and $v_-$ time dependent respectively (as Eq (5)), and the superscript $t$ identifies the model as a time dependent parameter model.

### 2.7.1 Inference of temporal dependence of mechanical parameters.

We fitted five different models with a single time dependent mechanical parameter, $M^t_{v_+}, M^t_{v_-}, M^t_\kappa, M^t_\alpha, M^t_\tau$, and four multiple time varying parameter models $M^t_{v_+ v_-}, M^t_{v_+ \tau}, M^t_{v_- \tau}, M^t_{v_+ v_- \tau}$, to each KT sister pair metaphase trajectory. The model relationship (nesting) is shown in Fig 3F. We used the same quality control criteria for trajectories as in the asymmetry model, see Methods. We calculated the Bayes factors to determine the significance of time dependence of the mechanical forces and the preferred model, see Methods. We find that a substantial proportion (46.6%) of kinetochores demonstrate significant time dependence (Fig 8A). Specifically, there is a proportion of sister pairs that show significant time dependence in $v_-$, 20.2% (pooling the models with time dependence in $v_-$, Table 4), $v_+$, 15.8%, $\tau$, 25.3%, $\alpha$, 6.5% and $\kappa$, 1%. This suggests that the K-fibre

**Table 3**. **K-fiber force variation within cells is larger than between cells.** Variance contributions of the biophysical parameters, grouping by cell and location (5 radial partitions) within the MPP. The total variance (residuals) is shown in column 2, and the between-cell and between-location variance are given in columns 3, 4 respectively. Statistically significant differences between groups at 1% significance is indicated by $^*$. The ratio of between cell to between location variance is given in column 5 and the size of effect (ratio of the population parameters' standard deviation to the population parameters' (absolute) mean) in column 6. Parameter are inferred on the model with asymmetry in $v_-$ and $v_+$, $M_{v\pm}^a$; the posterior median $v_-$ and $v_+$ values are averaged over the two sisters. Note: Some of the ANOVA assumptions are violated because of the low coverage of KT pairs per cell within each radius group. However, results on 3 radial boxes give similar results, see Table B in S1 Appendix, but again the results should be taken with caution. Data based $n = 383$ KT pairs from $N = 14$ cells; the reduced counts are due to the requirement that there are at least 3 sister pairs in each group.3

| Parameter | Variance | | | | |
| --- | --- | --- | --- | --- | --- |
|  | Total | Between-cell | Between-location | Between cell/location | total s.d/mean |
| $\alpha$ | $3.9 \times 10^{-5}$ | $2.6 \times 10^{-6*}$ | $3.8 \times 10^{-7}$ | 6.856 | 0.498 |
| $\kappa$ | $1.5 \times 10^{-4}$ | $2.2 \times 10^{-5*}$ | $1.8 \times 10^{-5*}$ | 1.177 | 0.563 |
| $L$ | $7.3 \times 10^{-4}$ | $1.2 \times 10^{-4*}$ | $2.3 \times 10^{-5*}$ | 5.270 | 0.033 |
| $\tau$ | $3.1 \times 10^{4}$ | $2.9 \times 10^{3*}$ | $1.6 \times 10^{4*}$ | 5.474 | 0.240 |
| $v_-$ | $1.8 \times 10^{-4}$ | $2.7 \times 10^{-5*}$ | $4.6 \times 10^{-5*}$ | 0.582 | 0.313 |
| $v_+$ | $2.8 \times 10^{-5}$ | $8.8 \times 10^{-6*}$ | $1.7 \times 10^{-6*}$ | 5.124 | 0.589 |
| $p_{incoh}$ | $1.1 \times 10^{-2}$ | $1.1 \times 10^{-3*}$ | $3.5 \times 10^{-3*}$ | 0.341 | 0.114 |
| $p_{coh}$ | $2.8 \times 10^{-4}$ | $5.1 \times 10^{-5*}$ | $3.0 \times 10^{-5*}$ | 1.725 | 0.018 |

pulling and pushing forces and the diffusive noise are time dependent. In contrast, few kinetochore pairs show significant time dependence in the centromeric spring strength $\kappa$ or centralising force $\alpha$, Table 4; therefore models with time dependent $\kappa$ or $\alpha$ were not explored further.

**2.7.2 K-fiber dynamic parameters tune towards an anaphase ready state.** To determine how the K-fibre biophysical properties vary in time, we analysed the posterior median of $v_{\pm,0}$, $v_{\pm,1}$ from model $M_{v\pm\tau}$ for all KTs. This model encompasses the main significant time dependence observed in our data, i.e., $M_{v_+}^t$, $M_{v_-}^t$, $M_\tau^t$, $M_{v_+v_-}^t$, $M_{v_+\tau}^t$, $M_{v_-\tau}^t$ are nested models of $M_{v\pm\tau}^t$ (Fig 3F). For trajectories that prefer a simpler nested model the inferred parameters were similar for $M_{v\pm\tau}$ and the nested model; in particular for any parameter without significant time dependence (thus preferring the nested model) its inferred rate of change on $M_{v\pm\tau}^t$ was consistent with zero, as indicated by the 95% credible intervals. The 3 parameters, $|v_-|, v_+, \tau$ have a negative correlation between the rate of change ($p_1$) and the magnitude of the initial value ($p_0$), (Fig 8C–8E), $p_{Corr} < 10^{-15}$ for all 3 correlations, with time dependent coefficients that are predominantly negative for $v_\pm$, 68.4 % with $v_{-1} < 0$, 70% with $v_{+1} < 0$, and positive for precision, 64.1% have $\tau_1 > 0$. This means that the strength of K-fiber pulling and pushing forces, and the diffusive noise typically decrease towards anaphase. Thus, K-fibers with higher magnitude pulling (or pushing) forces decrease faster, whilst the pulling, respectively pushing forces, of K-fibers with weak pulling, pushing are strengthened. This suggests that the forces are being focused (tuned) towards a set point. We call this set-point the *anaphase ready state*. At the anaphase ready state the K-fibers are time invariant, *i.e.,* when

**Table 4**. **Temporal dependence: Number of KT pairs (and percentage) that preferred each temporal model for the DMSO and nocodazole washout treated cells.** The total number of pairs per treatment is shown in the column header.

| Preferred Model | DMSO (n=788) | Nocodazole (n=762) | Totals (n=1550) |
| --- | --- | --- | --- |
| Time invariant | 421 (53.4%) | 294 (38.6%) | 715 (46.1%) |
| $\alpha$ | 51 (6.5%) | 20 (2.6%) | 71 (4.6%) |
| $\kappa$ | 6 (1%) | 12 (1.6%) | 18/1550 (1.1%) |
| $v_-$ | 103 (13%) | 92 (12.1%) | 195 (12.6%) |
| $v_+$ | 40 (5%) | 110 (14.4%) | 150 (9.6%) |
| $v_-, v_+$ | 42 (5.3%) | 101 (13.3%) | 143 (9.2%) |
| $\tau$ | 93 (11.8%) | 78 (10.2%) | 171 (11%) |
| $v_-\tau$ | 15 (1.9%) | 27 (3.5%) | 42 (2.7%) |
| $v_+\tau$ | 17 (2.1%) | 28 (3.7%) | 45 (2.9%) |
| $v_\pm\tau$ | 0 (0%) | 0 (0%) | 0 (0%) |

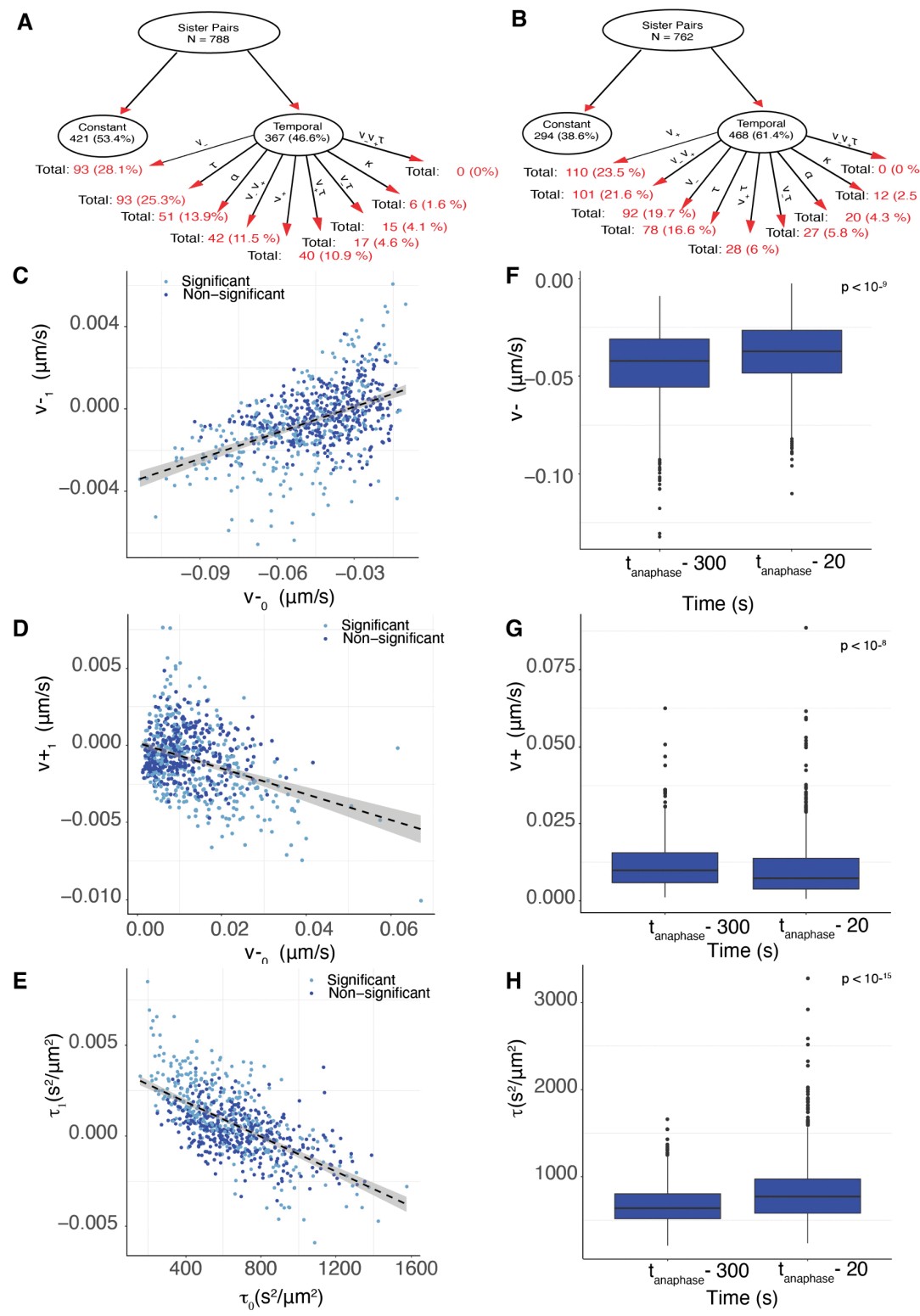

**Fig 8**. **K-fiber force strength decreases towards anaphase and are tuned towards an anaphase ready state (DMSO treated cells). A/B** Graphical representation of the temporal preference network with child node decomposition shown as percentage of the parent node for **A** DMSO and **B** nocoda-zole washout treated cells. **C/D/E** Scatter plot of posterior median of time dependence $p_1$ versus initial parameter value $p_0$ for **C** pushing forces, $v_{1,+}$

versus $v_{0,+}$; **D** pulling forces, $v_{1,-}$ versus $v_{0,-}$; **E** $\tau_{1,+}$ versus $\tau_{0,+}$. Light (dark) blue dots are KT pairs that have significant, respectively non-significant, time dependence. **F/G/H** Box-plots of posterior median parameter at the beginning and the end of the trajectory for **F** pushing forces $v_+$; **G** pulling forces $v_-$; and **H** noise $\tau$. Parameters values between mid- to late-metaphase are statistically different ($p_{MW} < 10^{-9}, p_{MW} < 10^{-8}, p_{MW} < 10^{-16}$ for $v_-, v_+, \tau$ respectively). Variances mid- to late-metaphase are statistically different with ($p_{BF} < 10^{-10}, p_{BF} < 0.02, p_{BF} < 10^{-21}$ for $v_-, v_+, \tau$ respectively). Parameters are inferred on the $M^t_{v_{\pm}\tau}$ model. Data in **A, C-H** from 788 KT pairs from 26 DMSO treated cells.

$v_{1-} = v_{1+} = 0$, giving the set point $v_{-*} = -0.033$ microns s$^{-1}(sd = 0.009)$, $v_{+*} = 0.003$ microns s$^{-1}(sd = 0.001)$ from the line of regression, Table 5. Tuning is observed in the majority of cells, for instance the same negative correlation between $v_{-0}$ and $|v_{-1}|$ is present in 26 out of 26 individual cells, with the majority (24 of 26) having a negative (and thus physical) intercept on the $v_{-1} = 0$ axis (Fig N in S1 Appendix). Tuning is also uniform across the MPP (Fig M in S1 Appendix). The anaphase ready state has lower (average) oscillation energy than during mid-metaphase, i.e.,, the forces from mid-metaphase reduce significantly over the 5 mins to anaphase (Fig 8F, 8G). This is consistent with the typically negative rate of change parameters (Fig 8). The KT pairs with significant time dependence are in fact those with initial values further from the anaphase ready state (Table 5), and thus have a larger rate of change, and a stronger time dependence. Their time dependence is therefore easier to detect.

To further explore tuning, we estimated for each KT pair the K-fiber forces $v_{\pm}(t)$ at 2 fixed reference time points, specifically at 300s prior to anaphase (in mid-metaphase) and 20s before anaphase onset (in late metaphase). The scatter plot (Fig K in S1 Appendix), clearly shows the focusing effect, with a fixed point at $v_- = -0.04$ microns/s (standard error (se) = 0.016 microns/s), $v_+ = 0.01$ microns/s (se = 0.009 microns/s), i.e., similar to the estimates of the anaphase ready state obtained above. These results suggest that $v_-$ has the following temporal dynamics,

$$\frac{dv_-}{dt} = (av_- + b)\, v_-,$$

since the average rate of change of $\log |v_-|$, $v_{1-}$ is correlated with $v_{0-}$, in fact giving $a, b > 0$, (Fig 8C). The general solution is:

$$\frac{1}{v_-(t)} = -\frac{a}{b} + e^{-bt}\left(\frac{1}{v_-(0)} + \frac{a}{b}\right),$$

with stable fixed point $v_- = -b/a$. This also defines a discrete map from $v_-(0)$ to $v_-(t)$ for a fixed $t$, as in Fig K of S1 Appendix, with a stable fixed point at $-b/a$.

As shown in Fig K(C) in S1 Appendix, the noise precision is also tuned, i.e.,the anaphase ready state has a set level of diffusive noise.

**Table 5**. **KT pairs with significant time dependence are those with initial values further from the anaphase ready state.** Posterior median estimates of $v_{\pm 0}$ for the significant and non-significant time dependent KT pairs, columns 2,3. Posterior distributions of these groups of KT pairs are statistically different using one-sided Kolmogorov-Smirnov (Mann-Whitney tests) for both $v_{\pm 0}$, column 5. Anaphase ready state estimates determined from regression of $v_{\pm 1}$ against $v_{\pm 0}$, column 4.

| | Posterior Medians | | | |
|---|---|---|---|---|
| | Time dependent KTs | Time invariant KTs | Anaphase ready state | p-value |
| $v_{-0}$ | −0.052 | −0.041 | $v_{-*} = -0.033$ | $< 10^{-8}$ ($< 10^{-7}$) |
| $v_{+0}$ | 0.012 | 0.009 | $v_{+*} = 0.003$ | $< 10^{-4}$ ($10^{-7}$) |

## 2.8 KT heterogeneity and trends are robust to perturbation of the spindle assembly pathway

To determine to what extent KT heterogeneity is sensitive to congression dynamics and conditions where segregation is more error prone, we disassembled MTs with nocadazole. The spindle reforms upon washout with microtubule nucleation at both centrosomes and at KTs, [49], and thus both the initial condition and spindle assembly dynamics are different. Congression may therefore utilise a different pathway. Although chromosomes congress and segregate under nocodazole washout treatment, segregation errors due to merotelic attachment are more frequent, [50,51], and there is a 3 fold increase in lazy kinetochores [52]. Lazy KTs are defined as KTs that significantly fall behind their cluster at any point during anaphase. At 4s time resolution, there were 0.26% lazy KTs in RPE1 cells, rising to 0.86% in nocodazole washout treated cells, [52]. It is unknown which of the mechanisms that result in the normally high fidelity of segregation are disrupted under nocodazole washout treatment.

We analysed KT heterogeneity in metaphase of nocodazole washout treated cells (330nM conc, 2 hr treatment followed by washout as in [52]), analysing 1240 KTs in 33 cells (out of 51 imaged cells using identical cell filtering requirements to previously). In these 33 cells detection and tracking performance was similar to cells cultured in DMSO; on average we obtained 89.9 KTs per frame, and 37.6 sister pairs per cell (quartiles Q1=35, Q3=41) where both sisters were tracked for at least 200 seconds (100 frames). The videos of nocodazole washout treated cells were typically longer than those of DMSO cultured cells, 237 frames versus 181 frames.

Metaphase dynamics is similar in nocodazole washout treated cells to DMSO cultured cells; the quasi-periodic oscillations had similar period and strength (ACF depth), with only weak variation across the MPP (Fig 9), that did not vary over the duration of metaphase (Fig C in S1 Appendix). KT oscillations were strongest in the middle of the MPP with a 20.2% (12.2%) increase in ACF depth at the MPP centre compared to the periphery in DMSO (nocodazole washout). The period was approximately constant across the MPP in both DMSO and nocodazole washout. MPP maturation over the last 5 mins of metaphase was substantially weaker in nocodazole washout although still significant, with a weak reduction in KK and the MPP width over the last 5 mins of metaphase (Fig C in S1 Appendix). This is possibly because the MPP width had already decreased prior to the 5 minute interval before anaphase, (Fig C in S1 Appendix), which contrasts to the steady reduction seen in DMSO (Fig 2). The average KK distance is marginally larger in nocodazole washout treated cells (1.135 microns compared to 1.104 microns, averaged over time and KTs ($p_{MW} < 0.005$), and there is a stronger spatial dependence across the MPP in nocodazole washout treated cells with a larger increase in the KK distance towards the periphery (Fig 9A).

We next determined if the heterogeneity in the biophysical parameters in time and space observed in DMSO are present in nocodazole washout treated cells. Most biophysical parameters were substantially changed in nocodazole washout treatment, however heterogeneity trends were essentially identical. Specifically:

1. **Biophysical parameters**. Nocodazole washout treated cells have on average weaker K-fibers, specifically the magnitude of $v_+, v_-$ were reduced by 37.5% ($p_{MW} < 10^{-15}$), and 5.8% ($p_{MW} < 10^{-4}$) respectively (Table G in S1 Appendix). They also have significantly reduced diffusive noise, ($p_{MW} < 10^{-15}$, precision increased by 20%), a higher centralising force ($p_{MW} < 10^{-10}$, $\alpha$ increased by 15%) and the spring constant is reduced by 40% ($p_{MW} < 10^{-9}$). The natural length is as expected the same as in DMSO, since the same informative prior is used. These data suggest that fewer MTs are bundled into K-fibers, leaving a higher density of spindle MTs that would explain the increase in the strength of the centralising force (increase in the PEF component).

2. **Spatial trends with the metaphase plate radius** were similar to DMSO (Figs 10 and I in S1 Appendix). In nocodazole washout treated cells, K-fibre pushing ($v_+$), pulling forces ($v_-$), the spring constant ($\kappa$) and diffusive noise ($\tau$) were reduced by 14%, 34%, 65%, 19% from the middle to the periphery of the MPP (in nocodazole washout treated cells). Finally, unlike DMSO, we observed a statistically significant reduction of 15% of the centralising force parameter $\alpha$ towards the periphery, ($p_{Corr}, p_{Kendall/Spearman} < 10^{-3}$).

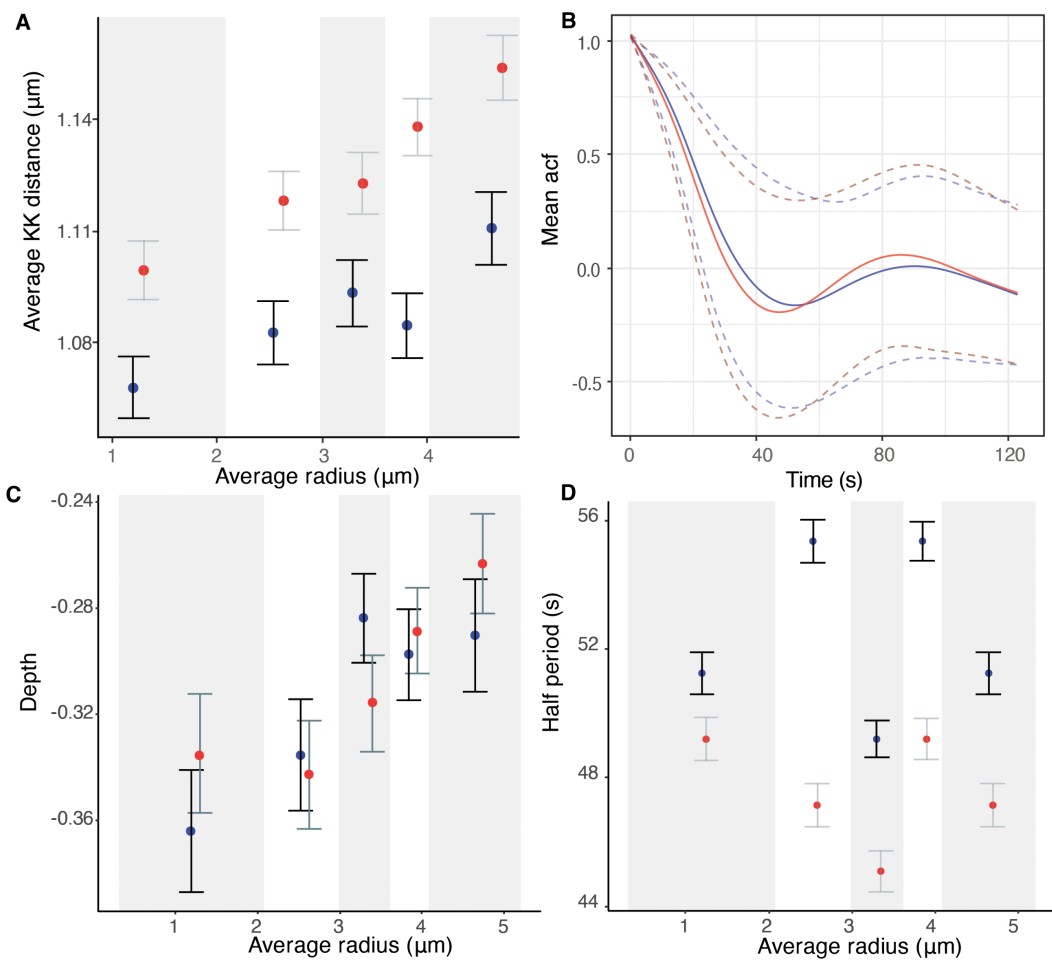

**Fig 9. Quasi-periodic oscillation period and strength vary weakly across the MPP and are robust to perturbation in DMSO (blue) and nocodazole washout (red) treated cells. A** Sister-sister separation (KK distance) against position in the MPP. **B** Mean ACF. Dashed lines denote the 5% and 95% percentile. **C** Average depth (ACF minimum) and **D** oscillation period for sister pairs across the MPP. Partitions are equally sized, containing the same number of observations (over pooled DMSO and nocodazole washout treated cells) The partitions are: [0, 2.07], (2.07, 2.97], (2.97, 3.59], (3.59, 4.01], (4.01, 5.20]. ACFs for each spatial partition are shown in Fig Y in S1 Appendix.

3. **Sister asymmetry** was similar with 28.5% having significant asymmetry (Fig 5B). There was an identical anti-correlation between the strength of pulling and pushing forces (Fig 5, and Fig O in S1 Appendix), and sister differences were identical to differences between random KTs. The lateral organisation transverse to the MPP into the strong/weak asymmetric KTs and symmetric KT groups was also present in nocodazole washout treated cells, (Fig 6B,D,F,H).

4. **Temporal trends** were very similar to DMSO (Fig 8B, Table 4) in fact there was a reduction in the preference for the time invariant model (53.4% DMSO to 46.1% nocodazole washout), likely due to the typically longer time series and the reduced diffusive noise under nocodazole washout, both increasing the ability to detect temporal variation. Significant temporal dependence was primarily seen in $v_{\pm}, \tau$ as before, with preferences for temporal dependence in $v_-$ (24.5%), $v_+$ (21.7%), $\tau$ (16.6%), whilst preference for temporal dependence in the other parameters is infrequent (Table 4). The correlation between the rate parameters and the initial parameter value was similar (Fig P in S1 Appendix), suggesting that K-fibers in nocodazole washout treated cells also tune towards an anaphase ready state.

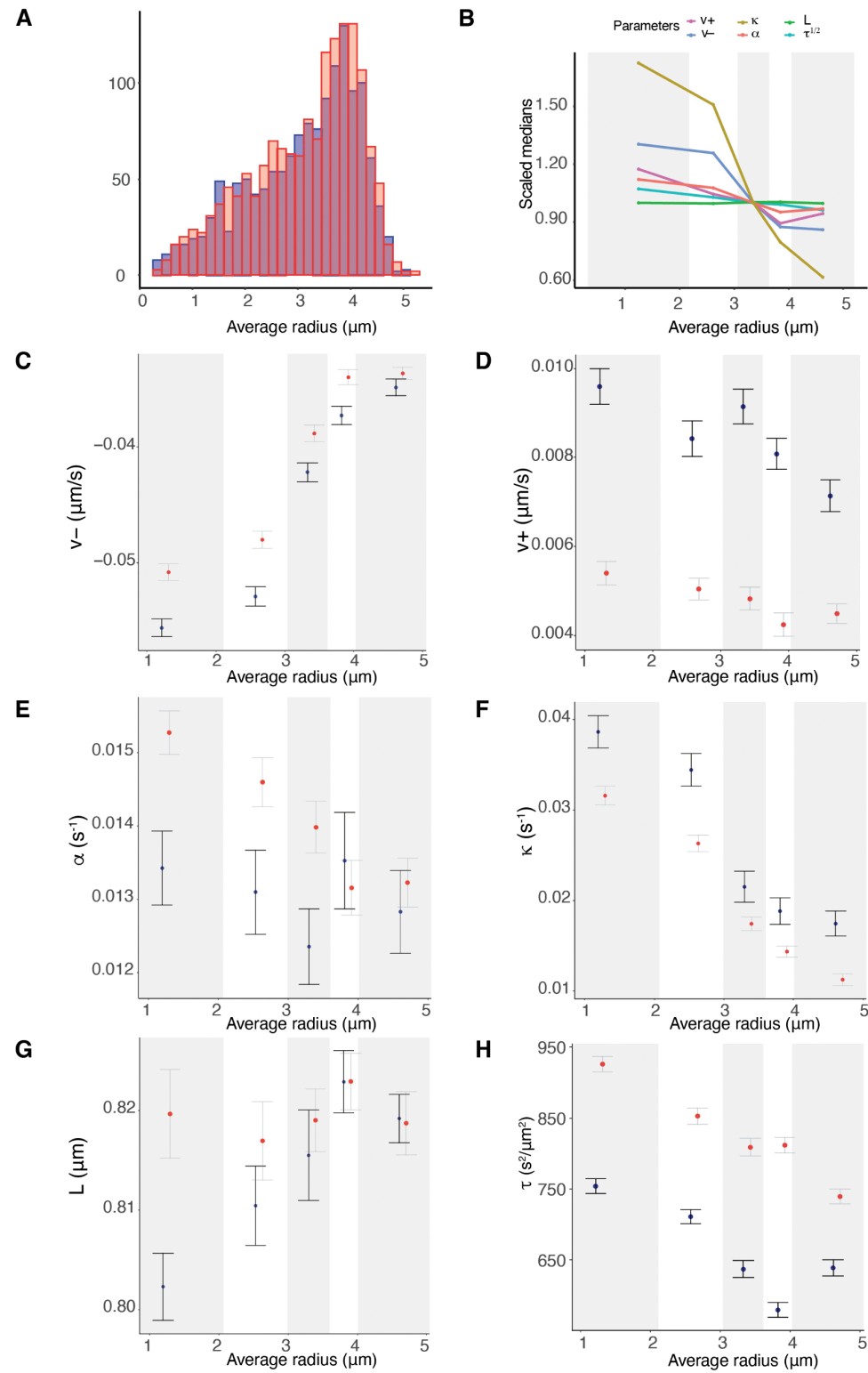

**Fig 10.** **Comparison of spatial mechanical parameter trends across the metaphase plate for DMSO (blue) and nocodazole washout (red) treated cells. A** Radial distance within the MPP of DMSO (blue) and nocodazole washout (red) treated KT pairs, based on 798 and 984 KT pairs respectively, over 59 cells. **B-F** Summary statistics (medians and standard deviations of posterior medians) of indicated biophysical parameters partitioned into 5 radial groups (average KT pair radial distance within the MPP). Partitions are equally sized, containing the same number of observations

(over pooled DMSO and nocodazole washout treated cells). The partitions are: $[0, 2.07], (2.07, 2.97], (2.97, 3.59], (3.59, 4.01], (4.01, 5.20]$. The partitions are not identical to that in Fig 7. **B** Summary plot of spatial trends over all biophysical parameters for nocodazole washout treated cells, similar to the summary for DMSO (Fig 7B). **C** Pulling forces $v_-$, **D** pushing forces $v_+$, **E** centralising force parameter $\alpha$, **F** spring constant $\kappa$, **G** natural length of the centromeric chromatin spring connecting the kinetochore sisters $L$, **H** precision $\tau$. Spatial trends for coherent and incoherent probabilities ($p_{coh}, p_{incoh}$) are shown in Fig I of S1 Appendix.

The anaphase ready state has slightly lower pulling force (magnitude) and pushing forces, (-0.02, 0.003) compared to (-0.04, 0.01) microns/s in DMSO, with this difference being statistically different ($p_{MW/Ztest} < 10^{-20}$). As before the K-fibre reduction in pulling/pushing force magnitudes over the last 5 mins of metaphase is significant ($p_{MW/KS} < 10^{-16}, p_{MW/KS} < 10^{-9}$ for $v_-, v_+$ respectively). There is little increase in the precision over that period (Figs L and P in S1 Appendix) whereas the precision increased in DMSO (Fig 8). This is possibly explained by the higher precision on all inferred models in nocodazole washout treated cells, for instance see (Fig L in S1 Appendix and compare Table 1 and Table D in S1 Appendix), and therefore any further increase may be limited. We confirmed that this higher precision isn't due to differences in movie length (reducing the length of movies had little effect on mean (median) estimates) as expected.

The heterogeneity in space, time and through random variation is thus robust to variation in spindle assembly dynamics, even under conditions with a higher chromosome mis-segregation rate, suggesting that this heterogeneity is intrinsic to mitosis, either tolerated or through design.

## 2.9 Oscillations are classified into 5 clusters varying in quality and strength and represent a distinct type of heterogeneity

Finally, we analysed the most obvious aspect of within cell variability – the fact that typical cells have a mixture of both good and poor oscillating KT pairs. Metaphase oscillations are a result of K-fiber forces and directional switching processes [22,26] and thus depend on the biophysical parameters. Therefore we address if the biophysical heterogeneity we documented above explains the variation in oscillation quality and strength, or if it represents another aspect of heterogeniety.

To analyse metaphase oscillation variability we clustered KT pair trajectories on the autocorrelation function (ACF) of the sister mid-point (mean of the sister $x$ coordinates) using hierarchical clustering with time warping, [53], (implemented in the "tsclust" R package [54]). We pooled the DMSO and nocodazole washout treated cell datasets to enable comparisons between treatments. The clustering dendrogram indicated that there were 5 clusters, (Fig 11A, 11B). Cluster signatures were fairly consistent across clustering methods. Visual inspection confirmed that there was consistency in the ACF of individual KT pairs within a cluster (Fig 11E–11I). The clusters had distinct average ACFs (Fig 11A) suggesting the following 5 phenotypes: *strong oscillators with short period* (ACF local maximum 89 s), *strong oscillators with longer period* (ACF local maximum 116 s), *weak oscillators with long period* (1/2 period of 60s), *weak and noisy oscillators* (1/2 period of 66s) and finally, *poor oscillators/non-oscillators*. Examples of trajectories in each category are shown in Fig S of S1 Appendix). The oscillatory category proportions are similar in DMSO and nocodazole washout treated cells (Fig 11C), with a slightly higher proportion of poor oscillators in DMSO treated cells, *i.e.,*35.8%, compared to 27.3% in nocodazole treated cells (Table H of S1 Appendix and Fig 11C). Cells varied substantially in the proportion of poor oscillators (Fig 11D) with 13.63% (6.45%) to 63.16% (46.88%) in DMSO (nocodazole washout) treated cells. Thus, some cells have poorer KT population oscillations than others, although each cell had a mixture of oscillators (Figs U and V of S1 Appendix).

To determine whether there are mechanical differences between our oscillation categories, we analysed the biophysical parameters of sister pairs in each cluster, using the parameter estimates of the asymmetric $M_{v_\pm}^a$ model. Strong oscillators

PLOS Computational Biology

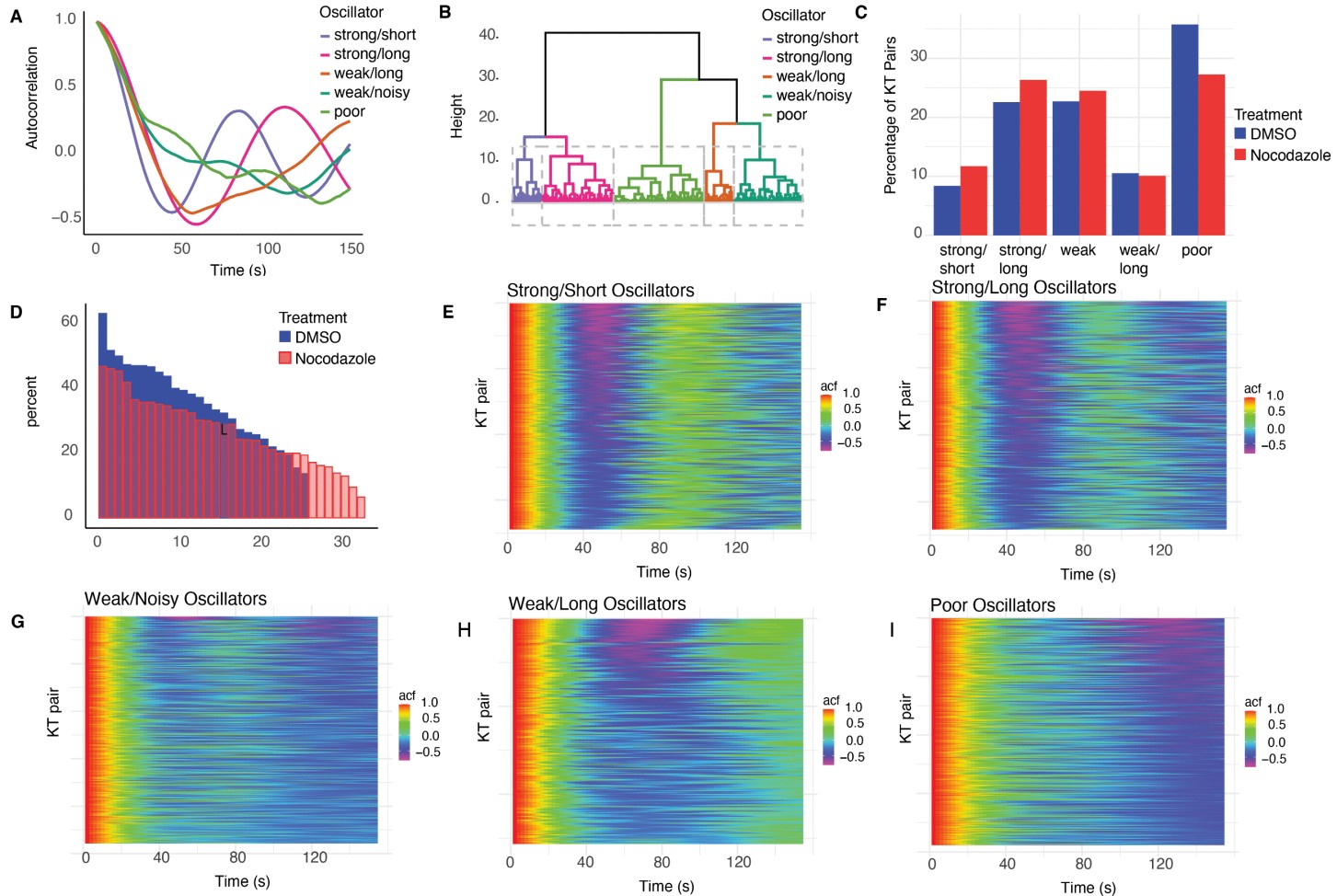

**Fig 11. Oscillator classification into 5 clusters - strong oscillators with short period, strong oscillators with long period, weak oscillators with very long period, weak and noisy oscillators and poor/non oscillators. A** Centroid ACF plots for each of the 5 clusters. Typical trajectories are plotted in Fig S of S1 Appendix. **B** Clustering dendogram (tree plot) showing cluster merger with distance/similarity level (height) for the hierarchical clustering. **C** Cluster proportions for DMSO (blue) and nocodazole washout (red) treated KT-pairs dataset. **D** Proportion of poor oscillators in each cell, for DMSO (blue) and nocodazole washout treatment (red) datasets (cells ordered by decreasing proportion of poor oscillators). **E-I** ACF of individual KT-pairs in each cluster, shown in pseudocolour. KT pairs stacked by row.

have higher K-fiber pulling forces, (higher $|v_-|$), and directional switching events take longer (lower probability of switching per frame $1 - p_{incoh}$) compared to weak and poor oscillators (Figs 12 and X of S1 Appendix). There is also a marginal decrease in the spring constant $\kappa$ and centralising force parameter $\alpha$ as oscillator quality decreases. The median parameter estimates were significantly different between the strong versus poor oscillators in most parameters, with $v_-, p_{incoh}$ having the highest significance, (Table I of S1 Appendix). There is a marginal decrease in the strength of the centromeric spring ($\kappa$) and in the strength of the centralising forces from strong to poor oscillators. This contrasts with oscillation amplitude that is reported to be dependent on the PEF, [55]. The period of the strong oscillator clusters is explained by $p_{coh}$, with the long period cluster having a higher probability of remaining in the coherent state, *i.e.,* higher $p_{coh}$ (Fig 12H).

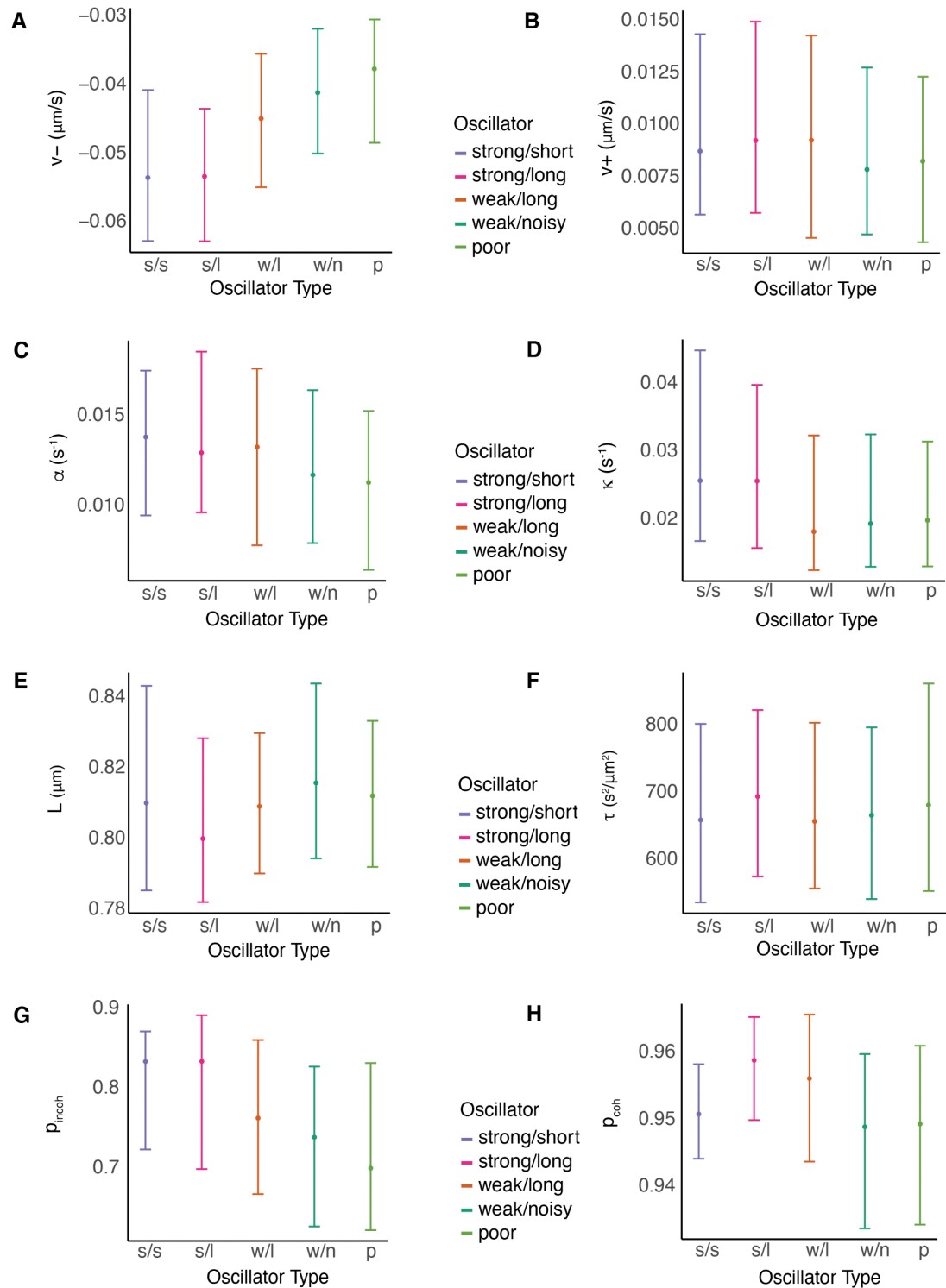

**Fig 12**. **Biophysical parameter variation with oscillation quality (DMSO).** Cluster median and standard deviation of stated parameter (posterior median) inferred on the asymmetric $M^{a}_{v\pm}$ model are plotted by oscillator type. Biophysical parameters weakly vary as oscillation quality decreases. The strongest trends are observed in the K-fiber pulling strength $|v_{-}|$ and the probability of remaining incoherent between frames ($p_{incoh}$), which decrease as the quality of oscillation deteriorates. Corresponding data for nocodazole washout treated cells is in Fig X in S1 Appendix. Posterior medians of $v_{\pm}$ inferred on the $M^{a}_{v\pm}$ model are averaged over sisters.

Oscillator type profile changed across the MPP, with the proportion of strong oscillators decreasing towards the periphery (Table H and Fig T of S1 Appendix) as previously reported on various cell types [17,18]. Since the biophysical parameters also have strong spatial trends (Fig 10), this change in oscillatory quality may be due to the changes in the biophysical parameters. We partitioned the KT pairs into 5 radial partitions (Fig W of S1 Appendix), and analysed $p_{incoh}$ and $v_-$ variation by radial partition and cluster. This demonstrated, as expected, that the strength of the pulling forces $|v_-|$ and the average duration of directional switching ($(1 - p_{incoh})^{-1}$ frames) decreased towards the periphery in each cluster (Fig 10), reflecting trends within the MPP of the KT population as a whole (Figs 10 and Fig I of S1 Appendix). However, within each spatial partition the strong oscillators have stronger pulling forces and longer directional switching times (higher $p_{incoh}$); in fact the directional switching time was significantly different between the strong and poor oscillators within most partitions for both DMSO and nocodazole washout treatments (Table J in S1 Appendix). A 2-way ANOVA analysis on $p_{incoh}$ with oscillator cluster (poor and strong only) and spatial partition indicates that 20% of variation is attributed to $r$, and 10% to oscillation cluster, with both radial distance and oscillation cluster being significant, ($p = 10^{-16}$). Similarly, for $v_-$, 16% is attributed to oscillation cluster and 18.4% is attributed to $r$, ($p = 10^{-16}$). There was no correlation between oscillation quality and asymmetry - the fraction of (significant) asymmetric pairs was not significantly different across oscillation clusters, $p_{CMH} = 0.41$ for DMSO treated cells and $p_{CMH} = 0.10$ for nocodazole washout treated data (test accounts for the distance from the centre of the metaphase plate). Thus, variation in oscillation quality is distinct from heterogeneity in the biophysical parameters.

## 3 Discussion

In this work we developed a data driven modeling framework for the study and analysis of KT dynamics at the cell level, revealing substantial heterogeneity in KT dynamics within a cell (Fig 13). Specifically, we observed both temporal organisation, with maturation of KT dynamics in time ($v_\pm, \tau$), thereby tuning system dynamics towards an anaphase ready state, and spatial organisation, with the MPP organised both within the MPP (trends in biophysical parameters $v_\pm, \kappa, \tau$) and transverse to the MPP (caused by random sister asymmetry in $v_-$). Metaphase oscillations are also heterogeneous, in part reflecting biophysical parameter trends within the MPP, but also representing a $4^{th}$ dimension of heterogeneity primarily through noise in directional switching control ($p_{incoh}$). Perturbing the spindle self-assembly pathway (nocodazole washout) gave similar results indicating that the processes governing heterogeneity and spatial-temporal trends are intrinsic to mitosis.

The majority of the parameter variation we report can be related to the K-fibers (Fig 13). Specifically, we observed (i) strong trends in the K-fiber force strength across the MPP, with both the pulling and pushing forces decreasing towards the periphery (38% and 25% respectively), (ii) asymmetry in the sister K-fiber pulling forces that results in transverse KT organisation across the MPP, and (iii) a decay of the pulling and pushing forces in the last 5 minutes of metaphase. MT polymerisation in K-fibers will likely contribute substantial active noise, [56], thus the spatial and temporal variation in diffusive noise probably reflects changes in MT dynamics and the cohesion of K-fibers, rather than a change in chromosome diffusion coefficients. What causes this spatial and temporal variation in K-fiber dynamics is unknown. We hypothesise that the observed temporal intrametaphase changes in K-fiber dynamics are due to regulatory processes that prepare the system for anaphase, primarily reducing the (kinetic) energy in the system. Intrametaphase molecular changes in the KT or K-fibers that could underpin these processes have not been reported, more likely reflecting the fact that quantification is difficult given the need to accurately time metaphase progression at high time resolution (secs) over a duration of minutes. In contrast, differences between prometaphase and metaphase have been reported [15,57]. As to spatial heterogeneity, there are known mechanisms that could generate such trends, specifically,

- The spindle is not a homogeneous environment, K-fiber and bridging fiber curvature increasing towards the periphery, [29,30,58]. This will change the geometry of the KT/K-fiber interface across the MPP. Bridging fibers are also under much higher compressive forces at the periphery compared to central K-fibers, [59] which may also impact KT dynamics

and biophysical properties. The spatial dependence of the K-fiber properties ($v_{\pm}$, $\tau$) within the MPP may therefore be an adaptation to these factors.

- Chromosome size and centromere size varies between chromosomes, and in particular, large chromosomes tend to be localised towards the periphery, [47,48]. Size could potentially impact a number of mechanical processes, and therefore be related to the within MPP trends we observe. Both centromere differences between chromosomes, [60], and KT size, [61], have been implicated in biasing chromosome segregation errors, in particular chromosomes 1 and 2 (the 2 largest chromosomes) have a higher mis-segregation rate, [51]. Size dependent mechanics may therefore impact segregation efficacy.

- Our analysis of sister asymmetry indicates that random variation contributes to KT heterogeneity within a cell, particularly in $v_{\pm}$, $\tau$, with variation in $v_{-}$ between sisters giving rise to transverse organisation of KTs in the MPP (Fig 13). This random variation in K-fiber properties could be a result of outer KT assembly at nuclear envelope breakdown (NEBD), [62], or the self-assembly of the K-fibers and spindle that occurs during prometaphase. Stochasticity in these self-assembly processes will introduce variability in K-fiber composition, bundle coherence, [63], retrograde flux and/or control processes. Since there is an anti-correlation between a K-fibre's pulling and pushing force strengths (Fig 5D), variation in K-fiber bundle size (MT count) is unlikely to be responsible since an increase in MT number within the bundle would likely increase the strength of both the pushing and pulling forces. Variation in K-fiber flux has the correct signature, *i.e.,* a decrease in flux increases pushing forces and decreases the pulling force strength. Further investigation is therefore needed to determine the cause of asymmetry and how MT bundle dynamics is controlled. Finally, the transverse organisation of the MPP may have consequences for segregation efficacy for affected chromosomes.

We also addressed if the spatial organisation in the K-fiber pulling forces in metaphase is preserved into anaphase. We fitted a metaphase-anaphase model to the trajectories, see Section D in S1 Appendix, and compared the K-fiber pulling force in metaphase with that in anaphase. There is a weak correlation between $v_{-}$ and the anaphase speed, $\rho = 0.174$, $p_{Corr} < 10^{-15}$, (Fig AB(F) of S1 Appendix), whilst spatial trends in the pulling forces within the MPP are lost in anaphase (Fig AB(C) of S1 Appendix) reminiscent of the anaphase speed governor, [64]. In fact, the anaphase speed of a KT pair had a higher correlation with its anaphase onset time, $\rho_{DMSO} = 0.247$ ($p_{Corr} < 10^{-9}$) in DMSO, $\rho_{noc} = 0.125$ ($p_{Corr} < 10^{-3}$) in nocodazole washout, (with $\rho_{noc} < \rho_{DMSO}$, $p_{Ztest} = 0.004$). The anaphase speeds were similar in DMSO and nocodazole washout ($p_{MW} = 0.083$), which contrasts to the lower pulling speed in nocodazole washout $p_{MW} < 10^{-4}$; pulling speed $|v_{-}|$ and anaphase speed were reduced by 5.85% and 2.8% respectively in nocodazole washout. This change in heterogeneity dependence likely reflects the fact that KTs are in different states in metaphase and anaphase. KTs undergo substantial phosphorylation at anaphase onset, [65,66], thus K-fiber dynamics will be regulated by different mechanisms in metaphase and anaphase.

We observed spatial trends in the centromeric spring constant $\kappa$, the stiffness decreasing 53% from the centre to the periphery of the MPP. However, our model assumed a linear spring, and therefore it is prudent to ask if nonlinearity in the spring, as reported in [15,67], could explain this trend. We observed that the spring extension (KK distance) increases towards the periphery (Figs 9 and 13), which contradicts the previously reported increase in stiffness with extension [15, 67]. Hence, nonlinearity is not the cause of the trends we observed in the spring constant with radial distance $r$. Centromere size could potentially affect the spring stiffness; however data on centromere size variation within the MPP isn't available. Since the change in $\kappa$ over the MPP is substantial a mechanism with a direct dependence on $r$ is suggested, for instance a consequence of the local geometry change across the spindle. The inferred spring constant in all our models, and in other studies such as [15], is an effective spring constant, so this must be considered as a possible cause. The centromeric spring models the behaviour of the connecting chromatic material between the sister chromatids, a composite material with likely anisotropic elasticity. There are two clear effects that might affect the effective spring constant upon moving towards the periphery. Firstly, a geometrical effect, K-fibers not being aligned with the spindle axis upon moving towards the periphery, [11] (Fig 13); thus a stretch of the sister-sister distance would then also comprise a shear and

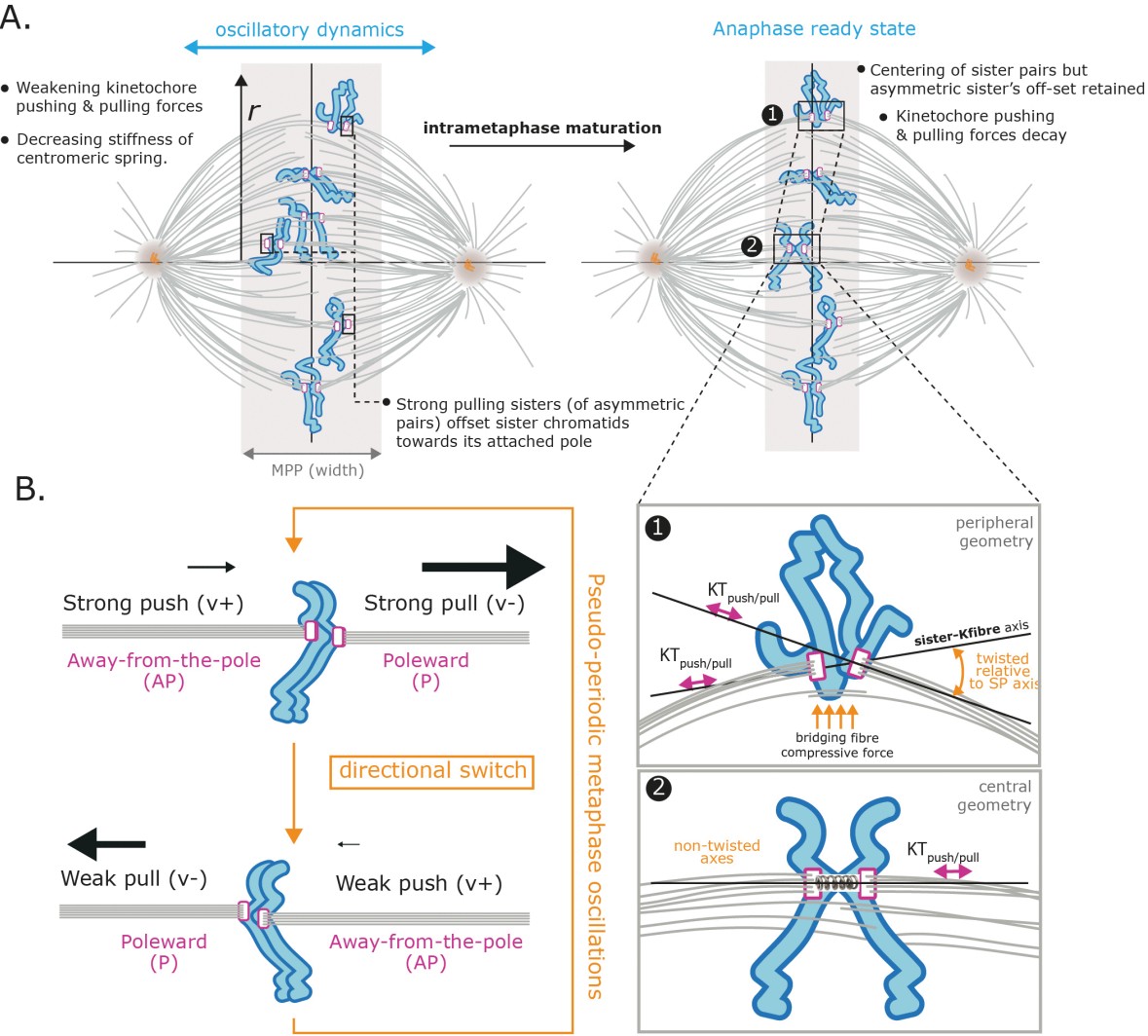

**Fig 13. Biophysical trends and organisation of KTs within the metaphase spindle. A** Left. Schematic of the spindle showing microtubules, bridging fibers, KTs (magenta) and chromosome arms (cyan). K-fiber pulling and pushing forces decay with radial distance ($r$), and the centromeric spring stiffness decreases. Strong pulling sisters of asymmetric pairs offset the sister pair towards its pole. Right. Late metaphase organisation after intrametaphase temporal tuning. The anaphase ready state has more centred chromosomes, and the K-fibre pulling and pushing forces typically decay. Sister chromatid offset is retained, but decreased. Annotated chromosomes at the periphery, (1), and central, (2), are shown in detail (lower panel), emphasising how the different local spindle geometry impacts attachment force orientation. **B** Schematic representation of forces in an asymmetric sister pair through the directional switching cycle (orange) of metaphase oscillations. Magnitude of the force is indicated by size of arrow. Schematic shows the inverse correlation between pulling and pushing strength of a K-fibre, whilst pulling forces are substantially larger than pushing forces.

torque. Secondly, chromosome arms take up a different geometry towards the periphery, being pushed outwards relative to the KTs, thereby generating a torque.

We identified two sources of random variation that impacts KT dynamics, random variation in K-fiber strength and random variation in directional switching parameters, the latter giving rise to variation in oscillator quality. Random variation in sister K-fiber strength, which was identical to differences between random KTs, resulted in transverse organisation of the MPP. In fact KT clusters that descend to respective poles in anaphase are organised laterally with KTs attached to weak pulling K-fibers on the inside, and KTs with strong pulling K-fibers on the outside (Figs 6 and 13). Crucially, this ordering

was preserved from mid to late metaphase in both DMSO and nocodazole washout treated cells, (Figs Q(D,E) and R(D,E) in S1 Appendix), suggesting that the tuning towards the anaphase ready state does not affect poleward bias of these groups. This lateral organisation is also preserved through to anaphase (Figs Q(F) and R(F) in S1 Appendix). Tuning was apparent in the descending clusters, with a statistically significant reduction in cluster spread from mid to late metaphase, *i.e.,* 0.468 to 0.414 microns in DMSO and from 0.351 to 0.321 in nocodazole washout treated cells ($p_{MW} < 10^{-15}$, one-sided for both treatments). Oscillations are also heterogeneous within cells, Section 2.9, with the proportion of poor oscillators varying substantially between cells. Strong oscillators have higher K-fiber pulling forces, higher $|v_-|$, and directional switching events take longer (higher $p_{incoh}$) compared to weak and poor oscillators (Figs 12 and X in S1 Appendix). As previously reported, [17,18], oscillation quality deteriorates towards the MPP periphery (Fig T in S1 Appendix), but despite these trends mirroring the trends in biophysical parameters with radial distance in the MPP, (Figs I and W(A,C) in S1 Appendix), they are not fully explained by location in the MPP. Thus, oscillation quality is another dimension to KT heterogeneity, distinct from spatial and temporal trends, and sister asymmetry.

Our analysis suggests that metaphase dynamics are robust to changes in the spindle environment. Specifically, despite the substantial variation in K-fibre length, curvature and structure (bridging fibers) across the MPP, [68], qualitatively similar dynamics occurs throughout the plate indicating mechanism robustness. In fact the strength of oscillations and the period only weakly vary across the MPP, (Fig 9). This robustness is of course already well documented; quasi-periodic oscillations are observed in other species and across different human cell lines, [7,13,17]. We examined robustness to spindle assembly dynamics using nocodazole washout treatment; oscillation quality was similar (Fig 11C) and we saw practically identical trends in biophysical parameters in space (within and transverse to the MPP) and in time (intrametaphase tuning), both effects also being observed in individual cells (Fig O in S1 Appendix). In nocodazole washout treated cells the centralising forces were stronger, K-fibers weaker (lower $|v_\pm|$), diffusive noise weaker and the anaphase ready state had lower energy (with lower $|v_{\pm*}|$). We hypothesise that K-fiber bundles are smaller in nocodazole washout, with fewer MTs, so are weaker but more coherent, thus reducing the active noise contribution to the diffusion.

We did observe distinct changes in anaphase dynamics under nocodazole washout treatment, suggestive of a decreased segregation efficacy. Firstly, on the descending clusters - there was a substantial increase in average cluster spread from 0.345 to 0.610 microns (*i.e.,* 77% increase) within the first 120s from anaphase onset (Fig Z(B) in S1 Appendix), whilst DMSO cluster size was invariant to a 17.5% tolerance (minimum to maximum mean cluster size). Secondly, we observed a higher frequency of transient direction reversal events, [5], (Fig Z in S1 Appendix). KTs with reversals had stronger pulling K-fibers (Table L, Figs AE and AF in S1 Appendix) consistent with a higher proportion of strong oscillators having reversals (Table K in S1 Appendix). Whether such reversals impact segregation efficacy is unknown.

Our data driven analysis suggests that human KT dynamics analysis should incorporate sister asymmetry and temporal variation. In particular, the following paired KT model is suggested,

$$X_{t+\Delta t}^k = X_t^k - \left( v_{\sigma_t^k,*} + \left( v_{\sigma_t^k,0}^k - v_{\sigma_t^k,*}^k \right) e^{v_{\sigma^k,1}^k t} \right) \Delta t - \kappa \Delta t \left( X_t^1 - X_t^2 - L\cos(\theta_t) \right)$$
$$- \alpha \Delta t X_t^k + \sqrt{\Delta t} N\left( 0, \left( \tau_0^k e^{-\tau_1^k t} \right)^{-1} \right),$$
(6)

where $X_t^k$ is displacement along the MPP normal for sister $k$, and $v_{\pm,*}$ is the anaphase ready state. Further work is required to confirm support for this joint asymmetric, temporal model, and determine whether it is identifiable, and in particular the requirements on time and spatial resolution for identifiability. There is also substantial variation in biophysical parameters with radial position in the MPP, so a full cell level model could be developed incorporating neighbouring KT interactions. However, we did not determine the functional form of this spatial dependence with respect to radial location $r$, which would be required for such a model. This could be inferred using Gaussian process models or splines.

Our framework utilising biophysical modelling within a Bayesian inference approach demonstrates that fundamental properties of KT biophysical behaviour can be extracted with simple models that reflect the data resolution and are thus identifiable. Although the models are simple, they capture the key mechanical processes, and thus have powerful knowledge generating capability. Further improvements are possible, both overcoming the simplifying assumptions of the current models and incorporating additional processes into the biophysical models to improve their realism, likely requiring additional datasets to infer those processes. Possible improvements include inference in 3D and utilising additional datasets that allow separation of the composite forces into their components. Specifically, our models are in 1D, with K-fiber forces assumed to lie along the axis. In practice, there is slight curvature of the trajectories in metaphase as they track the spindle geometry, so $v_{\pm}$ may be underestimated towards the periphery due to projection effects. The impact of bridging fibers on KT dynamics is also unknown; acute removal of PRC1, which weakens such fibers, does lead to changes in sister KT tension and orientation, [69]. Extending the models and analysis to 3D and incorporating bridging fibers, would thus capture these key difference between peripheral and central KTs. Our models are in terms of composite forces, specifically $v_{\pm}$ are the net K-fiber pushing (anti-poleward), pulling (poleward) forces respectively, comprising (de)polymerisation at the MT plus ends and retrograde flux (depolymerisation at the spindle pole). $\alpha$ models a linear centralising force comprising the PEF, flux driven centralisation, [16] and regulation of K-fiber plus end dynamics by HURP [45]. Deconvolving these composite forces, thus establishing their relative contributions will require additional experiments that perturb individual processes. The model for drag forces is also an approximation since the MT bundles exhibit high transverse drag, [70] and the spindle behaves as a visco-elastic anisotropic material, [71]. An anisotropic drag model is therefore suggested. Interactions between kinetochore pairs, [19] can also be incorporated into the models.

Here we have documented and quantified substantial within cell KT biomechanical heterogeneity in space, time and through random variation, a heterogeneity that is in fact observed in all the RPE1 mitotic cells analysed and exceeds biological variation between cells in many parameters (Table 3). It is unknown whether this heterogeneity is physiological, specifically if biophysical variability is required for efficient cell division. The fact that cancer cells have deteriorated metaphase oscillations and oscillation quality inversely correlates with the rate of chromosome instability [14], suggests that oscillation quality and variability may be important to cell division fidelity.

We suggest that the heterogeneity reported here is functional, reflecting both adaptation to cellular constraints (*i.e.,* geometry) and by design. Firstly, intrametaphase changes tune the KTs to an anaphase ready state, primarily reducing the energy in the oscillations that we hypothesise improves segregation efficacy. Secondly, we ascribe variation within the MPP as an adaptation to the local spindle environment, dictated by local spindle architecture and geometry. Thirdly, we propose that the transverse organisation of KT pairs in the MPP relieves steric constraints between chromatids as they align to the MPP. Finally, we propose that variability in oscillation quality and strength avoids oscillator synchrony. If all chromosomes oscillated in phase, strong periodic cytoplasmic flows would likely be generated that would increase viscous forces, increasing the risk of chromosome arm breakage at fragile sites. Such oscillations, and the associated hydrodynamic flows could also potentially generate strong forces within the spindle and impact spindle orientation processes. Addressing these questions will require data on targetted experimental perturbations. Such experiments will also allow determination of the impact of heterogeneity on segregation efficacy. Crucially our analysis emphasises that KT behaviour is highly variable within a cell implying that the dynamics of chromosomes in mitosis should be viewed in the context of the mechano-chemical machinery of the spindle rather than in isolation. This work thus sets the stage for future studies to quantitatively analyse the collective of KTs at the cell level.

## 4 Methods and materials

### Code

The trajectory data and original code used to generate the results reported in this work are available at Github, https://github.com/ckoki21/MetaAnaDynamics.git. Any additional information required to re-analyze the data reported in this paper is available from the lead contact upon request.

### Cell culture and generation of cell lines

A previously reported RPE1 cell line was used, [44]. In brief, an immortalized (hTERT) diploid human retinal pigment epithelial (RPE1) cell line (MC191), expressing endogenously tagged Ndc80-eGFP, was generated by CRISPR-Cas9 gene editing, [44]. hTERT-RPE1 cells were grown in DMEM/F-12 medium containing 10% fetal bovine serum (FBS), 2 mM L-glutamine, 100 U/ml penicillin and 100 mg/ml streptomycin (full growth medium); and were maintained at 37ºC with 5% $CO_2$ in a humidified incubator.

### Live cell imaging by lattice light sheet microscope

The lattice light sheet microscope (LLSM), [72], used in this study was manufactured by 3i (https://www.intelligent-imaging.com). Cells were seeded on 5 mm radius glass coverslips one day before imaging. On the imaging day, each coverslip was transferred to the LLSM bath filled with $CO_2$-independent L15 medium, where live imaging takes place. All imaged cells entered anaphase, which is a suitable proxy for a lack of phototoxicity effects, [7]. The LLSM light path was aligned at the beginning of every imaging session by performing beam alignment, dye alignment and bead alignment, followed by the acquisition of a bead image (at 488 nm channel) for measuring the experimental point spread function (PSF). This PSF image is later used for the deconvolution of images. 3D time-lapse images (movies) of Ndc80-eGFP were acquired at 488nm channel using 1% laser power, 20 ms exposure time/z-plane, 75 z-planes, 307 nm z-step and 0.5 s laser off time, which results in 2 s/z-stack time/frame. Acquired movies were de-skewed and cropped in XYZ and time, using Slidebook software in order to reduce the file size. Cropped movies were then saved as OME-TIFF files in ImageJ.

### Tracking

Kinetochore tracking is performed using the software package KiT v3.0. The tracking algorithm proceeds by detecting candidate spots via a constant false alarm rate (CFAR) detection method, [42], to set a KT-wise dynamic threshold per image frame in a movie. Candidate spot locations are refined by fitting a Gaussian mixture model. Spot locations are linked between frames by solving a linear assignment problem, with motion propagation via a Kalman filter. Tracked kinetochores are paired by solving another linear assignment problem. Sister kinetochore pairing used spatial and temporal trajectory data. Sister kinetochores are closer ($d_{ij}$, average distance) and exhibit less distance variation ($v_{ij}$, distance variance) than non sisters. In the presence of a metaphase plate, their connecting vector aligns with the plate's normal ($\alpha_{ij}$, average angle). Pairing costs were calculated as $d_{ij} \times v_{ij}$ without a plate or $d_{ij} \times v_{ij} \times \alpha_{ij}$ with a plate. Trajectories had to overlap for at least 10 frames, and the configuration with the minimum global cost was selected.

A quality control filter was applied to ensure good sister pair coverage per cell. Specifically, at least 30 sister pairs had to be tracked for 75% of the movie length, see Section C1 in S1 Appendix.

The code to perform kinetochore tracking is available from https://github.com/cmcb-warwick/KiT. The software includes a graphical user interface (GUI) for ease of use.

## Bayesian inference of biophysical parameters

We use a Bayesian approach to fit the biophysical models directly to each experimental sister pair trajectory, a method that infers all the biophysical model parameters jointly. In essence, Bayesian methods sample parameter values consistent with observed data. Crucially, this allows us to quantify uncertainty in the fitted parameters. We use the STAN programming language through the "rSTAN" package, [73,74] that implements an Hamiltonian Markov Chain Monte Carlo algorithm, [75,76]. This algorithm generates samples of the posterior parameter distribution $P(\theta|x_{1:T})$, specifically the distribution of the model parameters $\theta = (\tau, \alpha, \kappa, v_-, v_+, L, p_{coh}, p_{incoh})$ in the case of the model Eqs (1), (2), given a sister pair trajectory $x_t = [X_t^1, X_t^2]$. Details on the likelihoods for each model, how we deal with missing data and the posterior parameter and hidden state sampling (from the posterior distribution) are given in the Sections C2 and C3 of S1 Appendix. There is an identifiability issue in all models, specifically the centromeric spring's natural length $L$ is poorly inferred from the data. This is discussed in [17] and resolved by measuring the natural length in nocodazole (which depolymerises all MTs); the average natural length (KK distance) for RPE1 cells is 0.78 microns, similar to HeLa cells at 0.76 microns, [17]. All the models analysed in this paper then satisfy practical identifiablity [77,78].

We imposed additional tracking requirements for model inference, in particular trajectories have to have at least 120 time points and less than 20% missing data in metaphase, Section C1 in S1 Appendix. Inference was successful on the vast majority of trajectories; a small number of KT pairs ($\sim$1.5% in DMSO dataset) had severe divergences, likely due to the model being inappropriate, *i.e.,*not capturing the dynamics. Such trajectories were removed from the analysis. Hence, in the model analysis there are slight differences in the number of trajectories considered due to performance differences.

## Determining which models are supported by the data

Model selection is based on pairwise comparison of models using the Bayes factor. All model comparisons are nested; thus we assess the evidence for increasing the complexity of the model. The Bayes factor for model $M'$ relative to a simpler model $M$ is the fraction $B = \frac{\pi(D|M')}{\pi(D|M)}$ given data $D$; the model marginals $\pi(D \mid M)$ are computed using a bridging sampler, [79]. We use the four strengths of evidence criteria against the null hypothesis as defined by [41]; "Not worth more than a bear mention" *i.e.,BF*<3.2, "Substantial" *i.e.,*$3.2 \le BF < 10.0$, "Strong" *i.e.,*$10 < BF \le 100$, and "Decisive" , *i.e.,BF* $\ge$ 100. We only chose the more complex model if it had at least substantial preference over the simpler model (Bayes factor *BF*>3.2). We initially determine which models are supported relative to the vanilla model (Eq (1)), giving a set of preferred models. Within this set, any nested models were then assessed for support for an increase in the model complexity using the Bayes factor between pairs of models within the nesting (Fig 3E, 3F), removing the more complex models for which increased complexity is not supported. If multiple models remain, we select the model with the highest Bayes factor (relative to the vanilla model) as the preferred model. In this way we systematically navigate the model network to determine the model with the greatest support from the data, and with a model complexity justified by the data.

## Statistical tests

The statistical tests used in this study can be found in Table O in S1 Appendix.

## Acknowledgments

We gratefully acknowledge the initial work and software development by Jonathan Harrison. We are grateful to David Corcoran and to all the members of CAMDU for microscopy support.

## Supporting information

**S1 Appendix. Supplementary text, Supplementary Figures and Supplementary Tables.**
(PDF)

## Author contributions

**Conceptualization:** Andrew D. McAinsh, Nigel J. Burroughs.

**Data curation:** Constandina Koki, Alessio V. Inchingolo, Abdullahi Daniyan.

**Formal analysis:** Constandina Koki, Nigel J. Burroughs.

**Funding acquisition:** Andrew D. McAinsh, Nigel J. Burroughs.

**Investigation:** Constandina Koki, Nigel J. Burroughs.

**Methodology:** Constandina Koki, Nigel J. Burroughs.

**Project administration:** Andrew D. McAinsh, Nigel J. Burroughs.

**Resources:** Constandina Koki, Alessio V. Inchingolo, Abdullahi Daniyan.

**Software:** Constandina Koki, Alessio V. Inchingolo, Abdullahi Daniyan, Enyu Li.

**Supervision:** Andrew D. McAinsh, Nigel J. Burroughs.

**Validation:** Constandina Koki, Nigel J. Burroughs.

**Visualization:** Constandina Koki.

**Writing – original draft:** Constandina Koki, Nigel J. Burroughs.

**Writing – review & editing:** Constandina Koki, Andrew D. McAinsh, Nigel J. Burroughs.

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
