## [Decision Letter · Decision Letter 0]

24 Mar 2025

PCOMPBIOL-D-25-00039

Bayesian data driven modelling of kinetochore dynamics: space-time organisation of the human metaphase plate

PLOS Computational Biology

Dear Dr. Burroughs,

Thank you for submitting your manuscript to PLOS Computational Biology. After careful consideration, we feel that it has merit but does not fully meet PLOS Computational Biology's publication criteria as it currently stands. Therefore, we invite you to submit a revised version of the manuscript that addresses the points raised during the review process.

Please submit your revised manuscript within 60 days May 24 2025 11:59PM. If you will need more time than this to complete your revisions, please reply to this message or contact the journal office at ploscompbiol@plos.org. Please include the following items when submitting your revised manuscript:

We look forward to receiving your revised manuscript.

Kind regards,

Jing Chen

Academic Editor

PLOS Computational Biology

Jason Papin

Editor-in-Chief

PLOS Computational Biology

**Additional Editor Comments (if provided):**

While the reviewers have acknowledged the significance of the findings and methodology in this paper, both they and I find that the readability of the text, as well as the clarity and completeness of the results and methods, require improvement. In particular, the manuscript is difficult to read due to the frequent use of long sentences and convoluted sentence structures.

**Journal Requirements:**

**Reviewers' comments:**

Reviewer's Responses to Questions

**Comments to the Authors:**

Reviewer #1: The authors of this deep study use Bayesian inference to obtain parameters of a conceptually simple mechanical model of sister-kinetochore movements in metaphase and anaphase. The model posits that two sisters are connected by centromere elastic spring and are acted upon by the polar-ejection and K-fiber forces. The key to the interesting behavior of the model is that each of the two K-fibers can be in the (hidden) pushing and pulling state. The hidden states evolve as a discrete time Markov chain parametrized by the probabilities of a kinetochore remaining in the coherent (sisters move in the same direction) and incoherent (sisters move in opposite direction) states. The authors use beautiful data on kinetochore trajectories and data science methods to obtain all parameters of this model. They found that:

sister kinetochores exhibit substantial sister asymmetry in kinetochore forces;

symmetric and asymmetric sisters are organized transversely to the metaphase plate;

K-fiber parameters have substantial spatial variation across the metaphase plate and are time dependent, adjusting towards anaphase;

the parameters are robust to perturbation of the spindle assembly pathway.

Lastly, the metaphase model was extended to describe anaphase.

The study is a tour-de-force. It is a perfect example of how the models should be built from data.

One major problem is that the paper is extremely dense and hard to read. At its core is the jargony and impenetrable lists of numerical procedures that will be of little use for a general reader. I strongly recommend replacing these long technical passages with a common-sensical description of ideas behind the methods. Just describe in simple words how do you obtain parameters from measured trajectories, and refer to technicalities in the appendices.

Two minor things:

It is unclear what are the 18 models considered – needs clarification.

“This study provides unprecedented detail and analysis” – c’mon, guys – you’re Brits and should know that one does not praise oneself like that. Wait until other people say it about your study :)

Reviewer #2: I am uploading the review as an attachment.

Reviewer #3: Overview and recommendation

In this manuscript, the authors use Bayesian inference to obtain the parameters of a 1D model describing kinetochore dynamics in metaphase and anaphase. The authors analyze the experimental data using previously developed tracking techniques, which allowed them to study kinetochore dynamics in unprecedented detail. The authors extend the previously introduced 1D model by introducing asymmetry between sister kinetochores in a subset of parameters (pull/push force and noise precision), as well as time dependencies in another subset (pull/push force, noise precision, spring constant and polar ejection force). They decide to use the model for each individual kinetochore pair by estimating the Bayes factor for successive models of increasing complexity, obtained from the marginal distributions of the data conditioned by the model, and rejecting the complex model if it does not provide a significant improvement over the simpler one. The authors analyze data from RPE1 cells, in DMSO conditions and with nocodazole washout treatment. Since the parameters were derived for each kinetochore separately, the authors were able to access their spatial trends within the metaphase plate.

First, the asymmetry between sister kinetochores was assessed, which showed that the polymerization force was the key parameter responsible for sister asymmetry. The weaker pulling sister was generally centered behind the kinetochore cluster, and the stronger pulling sister was centered in front of the cluster, relative to its pole. They show that this positioning is conserved towards anaphase. They did not find any global asymmetry of the entire spindle, which could possibly arise from centrosome asymmetry.

The authors also show that, while centromeric spring and pulling force vary significantly within cells, polar ejection forces vary largely between cells. They found that their spatial dependence cannot be explained by variable drag, which could possibly arise from variable chromosome size.

Motivated by the observed maturation of the metaphase plate over time, the authors finally estimated the temporal trends of the model parameters. Their analysis shows that the pushing and pulling forces, as well as the noise precision, depend on time. Their main result is that the forces are tuned in time to the anaphase-ready state and that the noise decreases (the noise precision increases) as kinetochores approach anaphase.

The above analysis was repeated for cells treated with nocodazole washout, which yielded similar spatial and temporal trends compared with DMSO, as well as similar sister asymmetry. Surprisingly, the authors find heterogeneity in the speed of kinetochore clusters in anaphase for cells treated with nocodazole washout, as well as multiple reversals in their trajectory. Motivated by these observations, the authors developed a model of the anaphase transition that did not include either centromeric spring or polar ejection forces. From the inferred speed of anaphase and the time of start of anaphase, the authors find that sisters with a later transition to anaphase have a higher speed.

Overall, I find this manuscript exciting and methodologically sound, elucidating the role of k-fiber heterogeneity in space and time in a rigorously quantitative manner. The existence of such heterogeneity has been reported in previous studies, and the authors hypothesize that the observed temporal changes are part of control processes that prepare the cell for the upcoming anaphase. I believe that the authors' work will make a strong contribution to the field of cell division biophysics and inspire future theoretical and experimental work, and therefore I recommend it for publication. However, the readability of the figures and mathematical expressions in the manuscript could be significantly improved. I would also ask the authors to address six important issues, listed in the major revisions below.

1. Major points

1.1 The authors should include or at least discuss flux-driven kinetochore centering (Risteski, Cell Rep. 2022). Their main centering mechanism involves only the opposing polar ejection and pulling forces, but this description may not be complete.

1.2. Bridging microtubules are mentioned in the discussion as a possible cause of spatial variability in k-fiber-related parameters (pulling/pushing forces and noise precision), and the authors state that these properties adapt to the changing geometry at the metaphase plate periphery. A more detailed discussion is needed on how changing geometry propagates to k-fiber parameters.

1.3 The authors should state more clearly what exactly is the advantage of using Bayesian inference over extracting such parameters from experiments in a more direct way.

1.4 In section C (Methodology), I do not find an expression (or iterative formula) for the likelihood of the model parameters conditioned on the data, only the likelihood of the data conditioned on the model parameters. Since the former is used to calculate the median of the parameters, explicitly stating its formula is mandatory

1.5 The authors should discuss how merotelic attachments enter their model, since, as they mentioned, there is an increase in merotelic attachments in cells treated with nocodazole washout.

1.6 There are no results anywhere in the manuscript for the angle $theta$ between sister kinetochores. Is there a particular reason for omitting it? Also, no data about $p_icoh$ and $p_coh$ are shown, neither in the main figures nor in the supplement.

2. Minor points

2.1. It is not clear how the authors define the beginning of metaphase.

2.2. I suggest moving Figures 1C, 1E, 1F and 1H to the supplement. The authors should describe whether Figure 1D shows data for a single cell or if they are pooled across multiple cells.

2.3 The authors should better explain in the introduction why they would even consider more complex models of kinetochore oscillations and how this fits into existing research by other groups.

2.4 The labels on some plots are too small (e.g. Figs. 6C and 6D). Figure 5E looks stretched. The labels on the y-axis of the histogram are missing (e.g. Fig. 5E, Figs. 6A-F, 7A,10A,12E). Some plots are missing units (e.g. Figs. 8C-E).

2.5 In equation (1), $sqrt(\delta t)$ should be replaced by its inverse.

2.6. I would suggest moving the section on data quality control to Methods. (lines 190-201)

2.7. There seems to be an error in the caption of Figure 3. “The sawtooth oscillations arise because coherent states (++, −−) are of longer duration.”. I think it should be stated here (+-,-+).

2.8 The authors sometimes refer to the parameter $tau$ as noise, and sometimes as noise precision. I think the term noise precision should be used throughout the text.

2.9 In Table 2, it is unclear what the labels Significant, Strong, Decisive, Significant+. mean.

2.10 A short sentence should be added in the main text explaining how kinetochore pairing was determined, or at least the Methods should be cited.

2.11 I think there is an error in the order of the models in this sentence "In RPE1 cells, 24.4% of sister kinetochores have significant asymmetry, with the majority (42.6% of asymmetric pairs) preferring asymmetry in v− v+ while 87.7% have significant asymmetry in v−" (lines 263-265).

2.12 In Figure 6, the captions for Figures 6A-D are missing

2.13 In the second term in equation 3, the $\delta t$ term is missing

2.14 I believe, if scaling due to the drag coefficient is to be properly investigated, $tau^1/2$ should be used in Figure 7B, as already stated in the main text.

2.15 In equation (4) I think $tau$ should be replaced with $tau_0$.

2.16. The formulas below lines 409 and 411 should be enumerated.

2.17 Figures 8A and 8B are not consistent with Table 5. For example, 28.1% of the sisters in Figure 8A prefer the v-time dependence, while 13% of the sisters in Table 5 prefer this same dependence.

2.18 There is a typo in the caption of Figure 8, "time dependence of p1 relative to the initial value of parameter p1". The initial parameter should be labeled p0.

2.19 I suggest to include in the caption in Figures 8C-E what "significant" and "non-significant" means.

2.20 The points representing the data in Figure 9 should be of equal size. The caption in Figure 6B states that the gray lines show percentiles, but they are actually dashed lines.

2.21 I suggest to label the trajectory of the kinetochore pair in Figure 11C when a reversal event occurs.

2.22 $sqrt dt$ is missing in equation (5). I would suggest using $\delta t$ and $\delta X$ instead of $dt$ and $dx$ for consistency with other equations in the main text.

2.23 There is a typo in the matrix P(t) after line 511. qA is written without a time dependence in the fourth column.

2.24 I suspect that the v_anaphase and |v-| histograms were swapped in Fig. 12E, according to the main text, where it is stated that on average v_a is less than v-.

2.25 How are the radial margins of Figs. 12C and 12D defined, given the fact that the kinetochore clusters are moving?

2.26 I strongly recommend enumerating all equations in section C (Methodology)

2.27 Regarding the expressions after line 16 in section C: I assume that $tau^(n-1)$ and $2pi^-(n-1)$ should carry an additional power of ½ . I also think that the variances of the individual Gaussian factors should be $\delta t/tau$ instead of $1/tau$.

2.28 I think the expression for the log-likelihood (after Equation 9 in section C) is wrong, because there should be the logarithm of the sum instead of the sum only. I also don't see where these log-likelihoods are used.

2.29 After line 31 in section C, the expression for $\xi_{i,t} P (x1:t-1)$ has two repeated lines. After line 32, the first line of the expression $\xi_{i,t}$ seems incorrect, and this expression should start with the second line, which seems correct.

2.30 I'm not sure why the proportionality relations are given for the transition probability between hidden states, equation (10) in the methodology section. As I understand it, these transition probabilities need to be calculated in full form, and not only as proportionality relations, in order to calculate the probability of a particular sequence of hidden states.

**Have the authors made all data and (if applicable) computational code underlying the findings in their manuscript fully available?**

Reviewer #1: Yes

Reviewer #2: Yes

Reviewer #3: Yes

PLOS authors have the option to publish the peer review history of their article (what does this mean?). If published, this will include your full peer review and any attached files.

Reviewer #1: No

Reviewer #2: No

Reviewer #3: No

**Figure resubmission:**
---

## [Decision Letter · Decision Letter 1]

15 Aug 2025

PCOMPBIOL-D-25-00039R1

Bayesian data driven modelling of kinetochore dynamics: space-time organisation of the human metaphase plate

PLOS Computational Biology

Dear Dr. Burroughs,

Thank you for submitting your manuscript to PLOS Computational Biology. After careful consideration, we feel that it has merit but does not fully meet PLOS Computational Biology's publication criteria as it currently stands. Therefore, we invite you to submit a revised version of the manuscript that addresses the points raised during the review process.

Please submit your revised manuscript within 30 days Oct 15 2025 11:59PM. If you will need more time than this to complete your revisions, please reply to this message or contact the journal office at ploscompbiol@plos.org. Please include the following items when submitting your revised manuscript:

We look forward to receiving your revised manuscript.

Kind regards,

Jing Chen

Academic Editor

PLOS Computational Biology

Jason Papin

Editor-in-Chief

PLOS Computational Biology

**Additional Editor Comments (if provided):**

Please address the remaining minor comments raised by the reviewers, and make sure all the information is clear and consistent throughout the manuscript.

**Journal Requirements:**

**Reviewers' comments:**

Reviewer's Responses to Questions

**Comments to the Authors:**

Reviewer #1: I am satisfied with the revisions

Reviewer #2: The authors have carefully revised according to the review provided. I appreciate their efforts to improve the quality of the paper. I recommend the manuscript for publication. Below are a few minor comments:

1. The term 'viscose' should be changed to 'viscous' wherever applicable.

2. In the section in line 1076-1099, some sentences are dense and could benefit from restructuring for enhanced clarity.

Reviewer #3: The authors have significantly improved the readability and flow of the text, as well as the quality of figures. The core ideas of the manuscript are well arranged, and are easier to follow than in the original version. I find the introduction of oscillator quality as a 4th dimension for describing the kinetochore heterogeneity particularly appealing. I appreciate the authors' effort in developing and properly linking the multitude of imaging, statistical and numerical methods to reveal how the kinetochore movement is regulated within the metaphase of an individual cell. I also appreciate the inclusion of the newly introduced figure on the hierarchy of models, which makes their approach easier to grasp. There are still some minor issues I would ask the authors to address:

1. Minor issues

1.1 I still think Eq. (1) is inconsistent with Eqs. (3) and (4). In Eq. (1), the increment in time appears in the denominator on the left-hand side, and therefore there should be the inverse of the square root of the time increment multiplying the noise term on the right-hand side. What am I missing here?

1.2. Line 491 states that pushing and pulling forces between the asymmetric sisters are correlated: a weaker pushing force corresponds to a stronger pulling force and vice versa. This is evident from Fig. 5D. The term correlation is used in subsection 3.4 (line 492), whereas the term anti-correlation is used in subsection 3.8 (line 757) and in the Discussion (line 917). For clarity, I suggest using consistent terminology throughout the text.

1.3. In part C (Methodology), the symbol tau_0 appears alongside tau in the equation below line 73. Shouldn't only tau appear here, since there is no temporal dependence? It appears that there is a discrepancy in the exponents of tau_0 and 2 Pi between the equations below lines 73 and 75, respectively. It is unclear why there is the product over the index k in both of these equations, since they refer only to a single sister.

1.4 The definition of L(x) above line 88 seems inconsistent with L(x) above line 93 in the Methods section. Also, it seems that above line 88, it should be the logarithm of the sum, and not the sum of logarithms.

1.5 It seems that in the set of equations below line 92, part C, which calculates the forward probabilities, there should be P(x_{1:t-1})/P(x_{1:t}) instead of just P(x_{1:t-1}).

1.6 I do not understand the meaning of the symbol s_{t+i}, in the second of the set of equations below line 104. This symbol was not introduced before.

Overall I find the updated version well suited for publication in your journal, and I believe the results presented by the authors will significantly advance the understanding of how already identified mechanisms (microtubule dynamics, chromosome centering, and centromeric stiffness), together with their heterogeneity, give rise to the successful mitosis.

**Have the authors made all data and (if applicable) computational code underlying the findings in their manuscript fully available?**

Reviewer #1: Yes

Reviewer #2: Yes

Reviewer #3: Yes

PLOS authors have the option to publish the peer review history of their article (what does this mean?). If published, this will include your full peer review and any attached files.

Reviewer #1: No

Reviewer #2: No

Reviewer #3: No

**Figure resubmission:**
---

## [Editor Report · Decision Letter 2]

30 Sep 2025

PCOMPBIOL-D-25-00039R2

Bayesian data driven modelling of kinetochore dynamics: space-time organisation of the human metaphase plate

PLOS Computational Biology

Dear Dr. Burroughs,

Thank you for submitting your manuscript to PLOS Computational Biology. After careful consideration, we feel that it has merit but does not fully meet PLOS Computational Biology's publication criteria as it currently stands. Therefore, we invite you to submit a revised version of the manuscript that addresses the points raised during the review process.

Please submit your revised manuscript within 30 days Nov 30 2025 11:59PM. If you will need more time than this to complete your revisions, please reply to this message or contact the journal office at ploscompbiol@plos.org. Please include the following items when submitting your revised manuscript:

We look forward to receiving your revised manuscript.

Kind regards,

Jing Chen

Academic Editor

PLOS Computational Biology

Jason Papin

Editor-in-Chief

PLOS Computational Biology

**Additional Editor Comments (if provided):**

I appreciate the thorough effort you have made in addressing all the reviewers’ suggestions. However, I feel additional English language editing is needed to further enhance readability. This can be done either through a professional editing service or by using AI-based tools such as ChatGPT. Please also include a declaration of any such usage in the COI statement when submitting your revised manuscript. If AI tools are employed, kindly ensure that no unintended changes to the scientific meaning are introduced.

**Journal Requirements:**

**Reviewers' comments:**

**Figure resubmission:**
---

## [Editor Report · Decision Letter 3]

31 Dec 2025

Dear Prof Burroughs,

We are pleased to inform you that your manuscript 'Bayesian data driven modelling of kinetochore dynamics: space-time organisation of the human metaphase plate' has been provisionally accepted for publication in PLOS Computational Biology.

Best regards,

Jing Chen

Academic Editor

PLOS Computational Biology

Jason Papin

Editor-in-Chief

PLOS Computational Biology

---

## [Editor Report · Acceptance letter]

PCOMPBIOL-D-25-00039R3

Bayesian data driven modelling of kinetochore dynamics: space-time organisation of the human metaphase plate

Dear Dr Burroughs,

I am pleased to inform you that your manuscript has been formally accepted for publication in PLOS Computational Biology. Your manuscript is now with our production department and you will be notified of the publication date in due course.

With kind regards,

Anita Estes
